# Token-Level Self-Play with Importance-Aware Guidance for Large Language Models

**Tue Le**
Hanoi University of
Science and Technology
Hanoi, Vietnam
`tue.ldt210909@sis.hust.edu.vn`

**Hoang Tran Vuong**
Hanoi University of
Science and Technology
Hanoi, Vietnam
`hoang.tv224855@sis.hust.edu.vn`

**Quyen Tran**
Rutgers University
New Jersey, US
`qt60@rutgers.edu`

**Linh Ngo Van**
Hanoi University of
Science and Technology
Hanoi, Vietnam
`linhnv@soict.hust.edu.vn`

**Mehrtash Harandi**
Monash University
Clayton, VIC 3800, Australia
`Mehrtash.Harandi@monash.edu`

**Trung le**
Monash University
Clayton, VIC 3800, Australia
`trunglm@monash.edu`

## Abstract

Leveraging the power of Large Language Models (LLMs) through preference optimization is crucial for aligning model outputs with human values. Direct Preference Optimization (DPO) has recently emerged as a simple yet effective method by directly optimizing on preference data without the need for explicit reward models. However, DPO typically relies on human-labeled preference data, which can limit its scalability. Self-Play Fine-Tuning (SPIN) addresses this by allowing models to generate their own rejected samples, reducing the dependence on human annotations. Nevertheless, SPIN uniformly applies learning signals across all tokens, ignoring the fine-grained quality variations within responses. As the model improves, rejected samples increasingly contain high-quality tokens, making the uniform treatment of tokens suboptimal. In this paper, we propose SWIFT (Self-Play Weighted Fine-Tuning), a fine-grained self-refinement method that assigns token-level importance weights estimated from a stronger teacher model. Beyond alignment, we also demonstrate that SWIFT serves as an effective knowledge distillation strategy by using the teacher not for logits matching, but for reward-guided token weighting. Extensive experiments on diverse benchmarks and settings demonstrate that SWIFT consistently surpasses both existing alignment approaches and conventional knowledge distillation methods.

## 1 Introduction

Large Language Models (LLMs) [1, 2, 3, 4, 5] have demonstrated strong generalization across diverse tasks, including text summarization [6, 7], code generation [8, 9], and instruction following [10, 11]. However, they can also produce harmful content [12], hallucinate [13], or reinforce sociocultural biases [14, 11], underscoring the need for alignment with human values. To address this, Reinforcement Learning from Human Feedback (RLHF) [11] has emerged as the standard approach for preference

39th Conference on Neural Information Processing Systems (NeurIPS 2025).

alignment. RLHF typically involves collecting human feedback on model outputs to train a reward model, which is then used to fine-tune the base model using reinforcement learning techniques such as Proximal Policy Optimization (PPO) [15]. While effective, RLHF is complex and resource-intensive, requiring extensive high-quality human feedback. To address these challenges, Direct Preference Optimization (DPO) [16] simplifies the process by directly optimizing on preference data, bypassing explicit reward modeling. Nevertheless, DPO still depends heavily on costly human annotations, limiting scalability.

To better balance alignment performance and generation diversity by controlling the KL divergence at the token level, Token-level DPO (TDPO) [17] redefines the objective as maximizing restricted rewards in a sequential manner using the advantage function. Building on TDPO, TIS-DPO [18] argues that different tokens *should not be treated equally* and proposes principled methods for estimating token weights by contrasting two LLMs: one biased toward high-reward tokens and another favoring low-reward tokens. However, this approach requires training two additional LLMs, which can be prohibitively expensive in terms of computation.

To reduce reliance on costly human preference annotations, Self-Play Fine-Tuning (SPIN) [19] trains models to generate rejected responses using earlier versions of themselves. These are paired with human-annotated SFT samples to form synthetic preference pairs for iterative fine-tuning, eliminating the need for explicit reward models and preference data. However, SPIN applies uniform learning signals across all tokens, overlooking that both chosen and rejected responses may contain a mix of high- and low-quality tokens—weakening token-level gradients as the model improves. Technically, SPIN matches distributions through a self-play adversarial game to generate outputs close to ground-truth sentences, but this limits its ability to perform fine-grained, token-level alignment, which typically requires to leverage with the advantage function and token-based reward functions from reinforcement learning.

In this work, we adopt a token-level perspective for matching the distributions of interest, which naturally leads to the formulation of token-level reward functions. Notably, our approach to token-level distribution matching reveals a strong connection to the objective of maximizing the advantage function, thereby enabling the incorporation of token-level weighting. However, this formulation inherently resembles teacher forcing. To transform it into a token-level self-play framework, we recast the problem using the student forcing mechanism [20], ultimately deriving our final objective function. For the token weight estimation, we leverage a teacher model to provide fine-grained token-level reward signals, enabling the student model to focus more on informative tokens during learning.

In summary, we propose **SWIFT (Self-Play Weighted Fine-Tuning)**, an extension of SPIN that incorporates token-level importance weights estimated from a stronger teacher model. Unlike conventional distillation methods [21] that match logits or hidden states, SWIFT leverages fine-grained token-level rewards to guide the student model's attention toward more informative tokens. To address tokenizer mismatches between teacher and student, we introduce a general token mapping strategy based on shared surface segments, enabling accurate weight transfer without compromising alignment. While SWIFT benefits from access to a strong teacher, in scenarios where we do not have access to such a model, we observe that assigning uniform token weights or applying contrastive token importance estimation as proposed in [18] still provides meaningful improvements, as shown in our ablation study (Table 3). We validate SWIFT through extensive experiments under both alignment and knowledge distillation settings across multiple benchmarks, consistently demonstrating its effectiveness and efficiency in aligning LLMs with human preferences.

Our main contributions can be summarized as follows:

- We propose SWIFT, a fine-grained self-alignment and distillation method that leverages teacher models to provide token-level importance signals.

- We introduce a practical solution to handle tokenizer mismatches, enabling reliable token-level weight transfer between teacher and student models.

- We conduct comprehensive experiments on multiple benchmarks and settings, showing that SWIFT consistently outperforms existing alignment and distillation methods.

## 2 Preliminaries

### 2.1 Preference-Based Alignment in LLMs

Let $\pi_\theta$ denote a Large Language Model (LLM) parameterized by $\theta$. Preference-based alignment aims to adjust $\pi_\theta$ to prefer human-preferred responses over less desirable ones. Formally, we consider a dataset of triplets $\mathcal{D} = \{(x, y^w, y^l)\}$, where $x$ is the prompt, and $y^w$ and $y^l$ are the preferred (winning) and less preferred (losing) responses, respectively, such that $y^w \succ y^l$. The objective is to optimize $\pi_\theta$ such that it assigns higher likelihood to $y^w$ than $y^l$. Direct Preference Optimization (DPO) [16] addresses this by directly optimizing the model on preference data, bypassing explicit reward modeling. Despite its simplicity, DPO still requires high-quality preference annotations, which are costly to obtain.

### 2.2 Self-Play Fine-Tuning (SPIN)

To overcome the reliance on external annotations, Self-Play Fine-Tuning (SPIN) [19] introduces a self-play paradigm, enabling the model to iteratively refine itself without access to additional preference data. SPIN starts from a Supervised Fine-Tuned (SFT) model trained on a dataset $\mathcal{S} = \{(x, y)\}$, where $y$ is a high-quality human-annotated response to prompt $x$.

At each iteration $t$, the model $\pi_{\theta_t}$ generates synthetic responses $y'$ for prompts sampled from $\mathcal{S}$. These are paired with the corresponding human-annotated responses to create preference pairs $(x, y, y')$. The updated model $\pi_{\theta_{t+1}}$ is then trained to distinguish between these pairs, encouraging the model to prefer human-written responses over its own previous generations. Formally, SPIN optimizes the following objective:

$$\mathcal{L}_{\text{SPIN}}(\pi_{\theta_{t+1}}, \pi_{\theta_t}) = \mathbb{E}_{(x,y) \sim p_{\text{data}},\, y' \sim \pi_{\theta_t}(\cdot|x)} \left[ \ell \left( \beta \log \frac{\pi_{\theta_{t+1}}(y \mid x)}{\pi_{\theta_t}(y \mid x)} - \beta \log \frac{\pi_{\theta_{t+1}}(y' \mid x)}{\pi_{\theta_t}(y' \mid x)} \right) \right], \quad (1)$$

where $\ell(\cdot)$ is a convex decreasing loss function (e.g., logistic loss), and $\beta$ is a scaling factor.

While SPIN enables iterative self-improvement at the sequence level, it applies uniform learning signals across all tokens, overlooking token-level quality variations. As the model improves, this inefficiency grows, as rejected responses may still contain valuable tokens. To address this, we propose SWIFT, a token-level refinement framework that incorporates importance-aware token weighting.

## 3 Our Proposed Self-Play Weighted Fine-Tuning

### 3.1 Problem Setting

We are given a supervised training set $D = \{(x^i, y^i)\}_{i=1}^N$ where $x^i$ and $y^i$ are the input/output sequence of the tokens. Let $\theta^T$ denote the parameters of the teacher model, and $\theta^S$ the parameters of the student model to be trained. We denote the *student model at iteration $k$* as $\theta_k^S$. At the outset, we emphasize that the student model is primarily trained via weighted token-level self-play, with the teacher model providing guidance solely for estimating token importance.

We need to train the student LLM $\pi_{\theta^S}(y \mid x)$ to incorporate the knowledge from dataset $D$ and use the guidance from the teacher model for the token weight estimation. Let us denote $\mathbb{P}_d$ as the distribution of the ground-truth pair $(x, y)$, while $\mathbb{P}_{\theta^S}$ as the distribution of the synthetic pair $(x, y')$, where $y' \sim \pi_{\theta^S}(\cdot \mid x)$ with a ground-truth input $x$. Our task is to learn $\pi_{\theta^S}$ so that the distribution $\mathbb{P}_{\theta^S}$ remains closely related to the data distribution $\mathbb{P}_d$.

### 3.2 Our Proposed Approach

In what follows, we present the theoretical framework for our SWIFT. We denote $p_d(x, y)$ and $p_{\theta^S}(x, y)$ where $y = [y_1, \ldots, y_T]$ are the density functions of $\mathbb{P}_d$ and $\mathbb{P}_{\theta^S}$. The following lemma characterizes the necessary and sufficient conditions in order to $\mathbb{P}_d = \mathbb{P}_{\theta^S}$.

**Lemma 3.1.** *The necessary and sufficient conditions for $\mathbb{P}_d = \mathbb{P}_{\theta^S}$ are $p_d(y_t \mid x, y_{<t}) = p_{\theta^S}(y_t \mid x, y_{<t})$ for all $t \leq T$.*

We note that, similar to the distribution matching in SPIN [19], we assume that $y$ and $y'$ from $\mathbb{P}_d$ and $\mathbb{P}_{\theta^S}$ have the same length $T$ (e.g., the maximum sequence length). This is standard in sequential training, where padding is typically used to align sequences to the maximum length. Moreover, inscribed in Lemma 3.1, we need to estimate the divergence between $p_d(y_t \mid x, y_{<t})$ and $p_{\theta^S}(y_t \mid x, y_{<t})$. To this end, we examine the following Integral Probabilistic Metric (IPM) [22, 19]

$$\forall t : \max_{r_t \in \mathcal{R}_t^k} \mathbb{E}_{[x, y_{\leq t}] \sim D} \left[ r_t \left( [x, y_{<t}], y_t \right) - \mathbb{E}_{y'_t \sim \pi_{\theta_k^S}(\cdot \mid x, y_{<t})} \left[ r_t \left[ x, y_{<t} \right], y'_t \right] \right], \tag{2}$$

where $r_t \left( [x, y_{<t}], z \right)$ is the $t$-th token-based reward model belonging in a function family $\mathcal{R}_t^k$. Here we note that the index $k$ specifies the current iteration with the current student model $\theta_k^S$, while the index $t$ specifies the $t$-th token. We name $r_t \left( [x, y_{<t}], z \right)$ as a token-based reward function because it offers a *high reward value* for a ground-truth token $z = y_t$, while offering a *low reward value* for a synthetic token $z = y'_t$ with $y'_t \sim \pi_{\theta_k^S} \left( \cdot \mid x, y_{<t} \right)$.

Additionally, we can express the token-level reward model $[r_t]_t$ using a sequence-level reward function $r(x, \tilde{y})$ for any output sequence $\tilde{y}$, where each token-level reward is given by $r_t \left( [x, y_{<t}], y_t \right) = r \left( x, y_{\leq t} \right)$. To support variable-length inputs when computing reward outputs, we represent $r$ either with a dedicated transformer-based model or implicitly through our LLM, following approaches similar to DPO [16] or Self-Play [19]. We denote the function family for $r$ as $\mathcal{R}^k$. We assume the function family $\mathcal{R}^k$ is *consistent* with the token-based function families $[\mathcal{R}_t^k]_t$ as follows.

**Definition 3.2.** (**Consistency**) The function family $\mathcal{R}^k$ is said to be *consistent* with the token-level function families $[\mathcal{R}_t^k]_t$ if, for any sequence of functions $[r_t \in \mathcal{R}_t^k]_t$, there exists a function $r \in \mathcal{R}^k$ such that

$$r \left( x, y_{\leq t} \right) = r_t \left( [x, y_{<t}], y_t \right), \quad \forall [x, y_{<t}] \sim D.$$

Conversely, for any $r \in \mathcal{R}^k$, there exists a corresponding sequence $[r_t \in \mathcal{R}_t^k]_t$ that satisfies the same equality, ensuring bidirectional consistency.

In addition, we represent $r$ and $[r_t]_t$ as implicit functions parameterized by the LLM $\pi_\theta^S$, which is assumed to have infinite capacity (i.e., it can approximate any measurable function to arbitrary precision). As a result, the consistency property is inherently satisfied. Leveraging the consistency property, we can reformulate the original optimization problem (OP) in (2) into a more tractable form.

**Theorem 3.3.** *Assume the consistency property between $\mathcal{R}^k$ and $[\mathcal{R}_t^k]_t$, we can equivalently reformulate the original OP in (2) into the following OP*

$$\max_{r \in \mathcal{R}^k} \mathbb{E}_{(x, y) \sim D, y' \sim \pi_{\theta_k^S}(\cdot \mid x)} \left[ \sum_t \gamma^{t-1} r \left( [x, y_{<t}], y_t \right) - \sum_t \gamma^{t-1} r \left( [x, y_{<t}], y'_t \right) \right], \tag{3}$$

*where $\gamma \in [0, 1]$ is the discount factor.*

In particular, the OP in (3) aims to align $p_{\theta^S}(y_t \mid x, y_{<t})$ with $p_d(y_t \mid x, y_{<t})$ for all $t$, which in turn drives the joint distribution $p_{\theta^S}(x, y)$ to match the data distribution $p_d(x, y)$ for $(x, y) \sim D$. Additionally, the optimization problem in (3) follows the teacher-forcing paradigm. We can reformulate it into an equivalent student-forcing objective [20], which achieves the same goal of aligning the joint distribution $p_{\theta^S}(x, y)$ with the data distribution $p_d(x, y)$, as follows.

$$\max_{r \in \mathcal{R}^k} \mathbb{E}_{(x, y) \sim D, y' \sim \pi_{\theta_k^S}(\cdot \mid x)} \left[ \sum_t \gamma^{t-1} r \left( [x, y_{<t}], y_t \right) - \sum_t \gamma^{t-1} r \left( [x, y'_{<t}], y'_t \right) \right]. \tag{4}$$

Denote the reward function $R(x, y) = \sum_t \gamma^{t-1} r \left( [x, y_{<t}], y_t \right) = \sum_t \gamma^{t-1} r \left( x, y_{\leq t} \right)$ as the sum of the token-based reward functions, we can rewrite the OP in (4) as

$$\max_{r \in \mathcal{R}^k} \mathbb{E}_{(x, y) \sim D, y' \sim \pi_{\theta_k^S}(\cdot \mid x)} \left[ R(x, y) - R(x, y') \right]. \tag{5}$$

It is evident that (5) defines an IPM between the two distributions of interest, $p_d(x, y)$ and $p_{\theta^S}(x, y)$, which serves the same purpose as the objective in (3), as demonstrated in the following theorem.

**Theorem 3.4.** *Assume that the function class $\mathcal{R}^k$ is symmetric (i.e., if $r \in \mathcal{R}^k$, then $-r \in \mathcal{R}^k$), and that the distribution family $\{p_{\theta^S}(x, y) : \theta^S \in \Theta\}$ includes the data distribution $p_d(x, y)$. Under these conditions, optimizing the objective in either (3) or (5) to learn the student model $\theta^S$ is equivalent. Mathematically, the minimization over $\theta^S$ using (3) or (5) yields the same optimal solution.*

To generalize the optimization problem in (5), we consider a broader objective in the same spirit, formulated as follows:

$$\min_{r \in \mathcal{R}^k} \mathbb{E}_{(x,y) \sim D, y' \sim \pi_{\theta_k^S(\cdot|x)}} \left[ l \left( R\left(x, y\right) - R(x, y') \right) \right]. \tag{6}$$

where $l$ is a non-increasing function (e.g., $l(t) = \log(1 + \exp(-t))$).

Inspired by [17, 18], we consider the following alternative token-level objective function involving the advantage function:

$$\max_{\pi_{\theta^S}} \mathbb{E}_{x, y_{<t} \sim D, y_t' \sim \pi_{\theta^S}(\cdot | x, y_{<t})} \left[ \omega_t A_{\pi_{\theta_k^S}} \left([x, y_{<t}], y_t'\right) \right] - \beta D_{KL} \left( \pi_{\theta^S} \left( \cdot \mid [x, y_{<t}] \right) \| \pi_{\theta_k^S} \left( \cdot \mid [x, y_{<t}] \right) \right). \tag{7}$$

where the *advantage function* $A_\pi \left([x, y_{<t}], y_t'\right) := Q_\pi \left([x, y_{<t}], y_t'\right) - V_\pi \left([x, y_{<t}]\right)$ with the state-action value function $Q_\pi$ and state value function $V_\pi$, $D_{KL}$ is the KL divergence, and $[\omega_t]_t$ are the token weights representing the importance of the tokens.

We now explain why the objective function in (7) assists us in learning a good new student model $\pi_\theta^S$. Considering rolling out one-step for $V_\pi$, we can approximate

$$A_{\pi_{\theta_k^S}} \left([x, y_{<t}], y_t'\right) \approx r \left([x, y_{<t}], y_t'\right) - \mathbb{E}_{y' \sim \pi_{\theta_k^S}(\cdot | [x, y_{<t}])} \left[ r \left([x, y_{<t}], y'\right) \right] \tag{8}$$

Linking to the maximization in (7), by maximizing the advantage function, we effectively maximize $r \left([x, y_{<t}], y_t'\right)$, where $y_t' \sim \pi_{\theta^S}(\cdot \mid [x, y_{<t}])$. This encourages the student model to generate synthetic tokens with high reward values. From (2), this process implicitly pushes $p_{\theta^S}(\cdot \mid x, y_{<t})$ closer to $p_d(\cdot \mid x, y_{<t})$, as desired. Moreover, the second term, $\mathbb{E}_{y' \sim \pi_{\theta_k^S}(\cdot | x, y_{<t})} \left[ r \left([x, y_{<t}], y'\right) \right]$, is likely to remain moderately high due to the previous update, thereby exerting a stronger effect in guiding $p_{\theta^S}$ toward $p_d$.

Similar to [17, 18], we have the following lemma.

**Lemma 3.5.** *The optimization problem in (7) admits the following closed-form solution:*

$$\pi_{\theta^S}^* \left(z \mid x, y_{<t}\right) = \frac{\pi_{\theta_k^S} \left(z \mid x, y_{<t}\right) \exp \left\{ \frac{\omega_t}{\beta} Q_{\pi_{\theta_k^S}} \left([x, y_{<t}], z\right) \right\}}{Z \left([x, y_{<t}]; \omega_t, \beta\right)}, \tag{9}$$

*where $Z \left([x, y_{<t}]; \omega_t, \beta\right)$ is the partition function. Consequently, the Q-function can be expressed as:*

$$Q_{\pi_{\theta_k^S}} \left([x, y_{<t}], z\right) = \omega_t \beta \log \frac{\pi_{\theta^S}^* \left(z \mid x, y_{<t}\right)}{\pi_{\theta_k^S} \left(z \mid x, y_{<t}\right)} + \omega_t \beta \log Z \left([x, y_{<t}]; \omega_t, \beta\right). \tag{10}$$

The optimal solution in (9) and (10) hints us how to define the family function $\mathcal{R}^k$. Specifically, this is defined as

$$\mathcal{R}^k = \left\{ r : \exists \theta^S \in \Theta \land Q_{\pi_{\theta_k^S}} \left([x, y_{<t}], z\right) = \omega_t \beta \log \frac{\pi_{\theta^S} \left(z \mid x, y_{<t}\right)}{\pi_{\theta_k^S} \left(z \mid x, y_{<t}\right)} + \omega_t \beta \log Z \left([x, y_{<t}]; \beta\right) \right\}. \tag{11}$$

Certainly, the reward functions $r$, and hence $R$, are implicit functions of the policy $\pi_{\theta^S}$. To derive the objective for updating $\pi_{\theta^S}$, we need to substitute this dependency into (6). We now rewrite the objective function in (6) as presented in the following lemma.

**Lemma 3.6.** *The objective function in (6) can be further derived as presented in the following lemma.*

$$l \left(R\left(x, y\right) - R(x, y')\right) = l \left( \sum_t \gamma^{t-1} A_{\pi_{\theta^S}} \left([x, y_{<t}], y_t\right) - \sum_t \gamma^{t-1} A_{\pi_{\theta^S}} \left([x, y_{<t}'], y_t'\right) \right). \tag{12}$$

Based on Lemma 3.6, we can reformulate the OP of interest as shown in the following theorem.

**Theorem 3.7.** *The objective function of interest in (6) can be reformulated to*

$$l \left( u\left(x, y, y', \pi_{\theta^S}, \omega\right) - v\left(x, y, y', \pi_{\theta^S}, \omega\right) \right),$$

*where we have defined*

$$u\left(x, y, y', \pi_{\theta^S}, \omega\right) := \sum_t \frac{\beta}{\omega_t} \left[\log \frac{\pi_{\theta^S}\left(y_t \mid x, y_{<t}\right)}{\pi_{\theta_k^S}\left(y_t \mid x, y_{<t}\right)}\right] - \sum_t \frac{\beta}{\omega_t} \left[\log \frac{\pi_{\theta^S}\left(y'_t \mid x, y'_{<t}\right)}{\pi_{\theta_k^S}\left(y'_t \mid x, y'_{<t}\right)}\right],$$

$$v\left(x, y, y', \pi_{\theta^S}, \omega\right) := \beta D_{SeqKL}\left(x, y, \omega; \pi_{\theta^S} \| \pi_{\theta_k^S}\right) - \beta D_{SeqKL}\left(x, y', \omega; \pi_{\theta^S} \| \pi_{\theta_k^S}\right)$$

*with the weighted sequence KL divergence being defined*

$$D_{SeqKL}\left(x, y, \omega; \pi_1 \| \pi_2\right) := \sum_t \omega_t^{-1} D_{KL}\left(\pi_1\left(\cdot \mid x, y_{<t}\right) \| \pi_2\left(\cdot \mid x, y_{<t}\right)\right).$$

Finally, the objective function to train our approach is

$$\min_{\pi_{\theta^S}} \mathbb{E}_{(x,y)\sim D, y'\sim\pi_{\theta_k^S}(\cdot|x,y)} \left[l\left(u\left(x, y, y', \pi_{\theta^S}, \omega\right) - v\left(x, y, y', \pi_{\theta^S}, \omega\right)\right)\right], \tag{13}$$

where $l\left(t\right) = \log\left(1 + \exp\left(-t\right)\right)$ is a non-increasing function.

Pseudo-code for our method can be found in Appendix B.4 in the supplementary material.

### 3.3 Teacher-Guided Token Importance Estimation

Our aim is to develop an efficient mechanism to estimate the importance weight of each token. Intuitively, a well-trained teacher—having been trained on more data with higher model capacity—can better signal which tokens carry higher "reward". In this section, we present a principled method to distill token-level reward signals from a teacher model $\theta^T$ and transfer them to a student model $\theta^S$. Due to space limitations, we leave a detailed explanation of this mechanism in Appendix B.5.

Inspired by TIS-DPO [18], we first estimate the raw importance weight for the $t^{\text{th}}$ token in a response $y = (y_1, \ldots, y_T)$ as

$$\omega_t^{\text{raw}} = k \cdot \exp\left(\mu \cdot \text{clamp}\left(\log \frac{\pi_{\theta^T}\left(y_t \mid x, y^{<t}\right)}{\pi_{\theta^S}\left(y_t \mid x, y^{<t}\right)}, L, U\right)\right), \tag{14}$$

where $\text{clamp}(a, L, U) = \min\left(\max(a, L), U\right)$ truncates the log-odds to $[L, U]$ to reduce variance and stabilize optimization. Here $k > 0$ and $\mu$ are constant; $L$ and $U$ are lower and upper clipping bounds.

However, this formulation assumes that the teacher and student models share the same tokenizer and vocabulary. In practice, this assumption rarely holds. A sequence $y$ might be tokenized by the teacher model as $[y_1^T, y_2^T, \ldots, y_t^T]$, and by the student as a different sequence $[y_1^S, y_2^S, \ldots, y_s^S]$, making direct token-to-token alignment in (14) infeasible. To address the misalignment issue caused by differing tokenizers, we propose a generalizable token mapping strategy based on shared surface segments.

**Case 1: Shared Tokenizer.** In the scenario where both the teacher and student models utilize an identical tokenizer, the alignment between their respective token sequences is inherently preserved. Specifically, each token $y_t$ generated by the student model corresponds directly to a token produced by the teacher model at the same position, thereby eliminating the need for additional alignment mechanisms. As a result, the distillation weight $\omega_t$ associated with each token position $t$ can be directly adopted from the unprocessed or raw weighting scheme. Formally, we define:

$$\omega_t = \omega_t^{\text{raw}}, \tag{15}$$

where $\omega_t^{\text{raw}}$ represents the original distillation weight derived from the teacher model's outputs as in Eq 14 without any post-processing or re-alignment.

**Case 2: Different Tokenizers.** When tokenizers differ, direct index alignment between tokens becomes unreliable. Rather than aligning tokens individually, we instead leverage a common structural unit shared across both tokenizations. Most modern tokenization algorithms, including BPE [23], SentencePiece [24], and WordPiece [25], segment text into subwords while generally respecting word boundaries. This means that each token strictly belongs to either a single word

or the whitespace separating words, making word spans a natural and reliable basis for alignment. Leveraging this property, we segment the original text into coarse-grained units corresponding to **complete words** and **their leading whitespace**, resulting in a sequence of shared components $\{c_1, c_2, \ldots, c_K\}$. This ensures that every token from both teacher and student sequences can be assigned to one of these shared components, which are small enough to preserve local semantics but broad enough to encompass all subword variations.

Suppose student token $y_i^S$ belongs to component $c_h$, and the teacher tokens in that segment are $T(c_h) = \{y_j^T, y_{j+1}^T, \ldots, y_{j+\ell}^T\}$. Assuming all tokens within a shared component contribute equally, we define the student's token importance as the average of the teacher tokens' weights in the corresponding component $c_h$:

$$\omega_i^S = \frac{1}{|T(c_h)|} \sum_{y_r^T \in T(c_h)} k \cdot \exp\left(\mu \cdot \text{clamp}\left(\log \frac{\pi_{\theta^T}(y_r^T \mid x, y_{<r}^T)}{\pi_{\theta^S}(y_i^S \mid x, y_{<i}^S)}, L, U\right)\right). \tag{16}$$

**Comparison with Prior Work.** Compared to the contrastive-based token weighting in [18], our method is significantly more computationally efficient, as it does not require training separate positive and negative models. We validate this by providing a direct comparison of the total time cost between SWIFT and the contrastive-based token weighting approach (TIS-DPO) in Appendix C.1 in the supplementary material. Furthermore, as shown in Table 3, our teacher-guided estimation consistently achieves superior performance, confirming both its efficacy and practicality.

## 4 Experiments

We evaluate SWIFT through extensive experiments across diverse settings, highlighting several key findings. (1) SWIFT consistently outperforms existing alignment methods on multiple benchmarks. (2) Since SWIFT relies only on an SFT dataset without preference-labeled data, we also compare it with knowledge distillation baselines, where it similarly outperforms, demonstrating the effectiveness of token-level importance estimation for distilling teacher knowledge. (3) Furthermore, ablation studies confirm that using the teacher to estimate token weights yields significant gains over alternative weighting methods.

### 4.1 Experiment Setup

We conduct two experimental settings to evaluate SWIFT: **Alignment** and **Knowledge Distillation**. The Alignment setting compares SWIFT with existing alignment methods, while the Knowledge Distillation setting assesses its effectiveness against existing knowledge distillation methods.

For the **Alignment** setting, We use Qwen1.5-1.8B [26] as the base model. As the teacher model, we adopt Zephyr-7B-SFT-Full [27], which is based on Mistral-7B [28] and further fine-tuned on the `Ultrachat200k` dataset[1] provided by HuggingFace. We follow the procedure in [19], 50,000 prompts are randomly sampled from `Ultrachat200k` and generate synthetic responses using the base model. For evaluation, we adopt the HuggingFace Open LLM Leaderboard [29], which is commonly used to assess the underlying capabilities of models through few-shot evaluation. Additionally, we evaluated the output quality of the LLM using MT-bench [30] with its provided dataset, we use the API of GPT-4 as the judge.

For the **Knowledge Distillation** setting. We use GPT2-1.5B [31] as the base model and Qwen2.5-7B-Instruct [32] as the teacher. Four datasets are selected for evaluation: `DATABRICKSDOLLY-15K` (**Dolly**) [33], `ALPACA` (**Alpaca**) [34], `S-NI` (**S-NI**) [35], and `DIALOGSUM` (**Dialogsum**) [36]. Final performance is reported using the ROUGE-L metric [37] between generated outputs and human-annotated references.

We perform 4 iterations, each iteration consisting of 2 epochs of training. Additional implementation details are provided in Appendix B in the supplementary material.

---

[1] `https://huggingface.co/datasets/HuggingFaceH4/ultrachat_200k`

Table 1: Performance of SWIFT based on Qwen1.5-1.8B across HuggingFace Open LLM Leaderboard datasets, compared with all baselines. * Methods trained on `Ultrachat200k` SFT data. †Methods trained on UltraFeedback Binarized preference data [39].

| Methods | Arc | TruthfulQA | Winogrande | GSM8k | MMLU | HellaSwag | Avg |
|---|---|---|---|---|---|---|---|
| Teacher | 60.41 | 43.73 | 74.19 | 26.76 | 60.92 | 82.85 | 58.14 |
| SFT* | 39.08 | 38.42 | 58.64 | 19.03 | 41.30 | 60.09 | 42.76 |
| DPO† | 39.33 | 38.37 | 59.12 | 19.18 | 41.30 | **61.74** | 43.17 |
| IPO† | 37.29 | 38.09 | 61.04 | 32.52 | 44.40 | 60.62 | 45.66 |
| TDPO† | 39.08 | 38.12 | 58.47 | 20.70 | 41.29 | 61.22 | 43.15 |
| TIS-DPO† | **40.44** | 39.09 | 61.27 | 30.71 | 41.61 | 61.66 | 45.80 |
| SPIN* | 39.93 | **40.46** | 58.17 | 18.42 | 40.81 | 61.42 | 43.20 |
| **SWIFT*** | 39.78 | 39.12 | **61.48** | **37.93** | **44.84** | 61.63 | **47.46** |

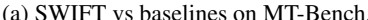

(a) SWIFT vs baselines on MT-Bench.      (b) Training reward comparison.

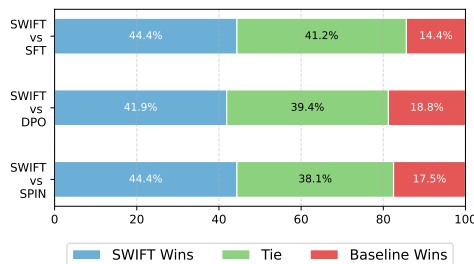
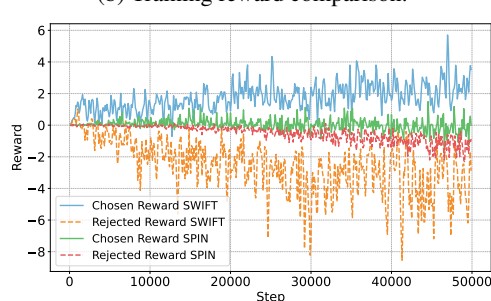

Figure 1: The left figure compares SWIFT with SFT, DPO, and SPIN on MT-Bench, evaluated by GPT-4. The right figure shows the trends of chosen and rejected rewards for SWIFT and SPIN during iteration 0.

## 4.2 Main Results

### 4.2.1 Comparison with Alignment Methods

Table 1 compares SWIFT with SFT and alignment baselines, including DPO [16], IPO [38], TDPO [17], TIS-DPO [18], and SPIN [19]. Following [19], DPO, IPO, TDPO, and TIS-DPO are trained on the UltraFeedback Binarized dataset [39], which contains 62k GPT-4-labeled preference samples, incurring significant data collection costs. Across six Open LLM Leaderboard benchmarks, SWIFT achieves the highest average score (47.46), surpassing the strong baseline TIS-DPO by a margin of +1.66, with strong gains on GSM8k (+7.22) and MMLU (+4.03) over SPIN. Complementing the leaderboard results, MT-Bench pairwise comparisons with GPT-4 as the judge (Figure 1a) show SWIFT consistently outperforming SFT, DPO, and SPIN, achieving win rates of 44.4% against SPIN and SFT, and 41.9% against DPO, confirming its advantage in response quality.

Figure 2 further supports this by illustrating performance trends across multiple iterations. While both SWIFT and SPIN start from the same SFT baseline, SWIFT demonstrates a clear upward trajectory across iterations, especially on MMLU and HellaSwag.

### 4.2.2 Evaluation against Knowledge Distillation Baselines

To further validate the effectiveness of our proposed method, we evaluate SWIFT under a knowledge distillation (KD) setting, where a smaller student model is distilled from a larger teacher model. As shown in Table 2, we compare SWIFT against Supervised Fine Tuning (SFT) and several existing KD baselines, including ULD [40], MinED [41], DSKD [42]. The teacher model is Qwen2.5-7B-Instruct, while the student is GPT2-1.5B. Across all benchmarks, SWIFT achieves the highest average ROUGE-L score (29.20), consistently outperforming previous distillation methods.

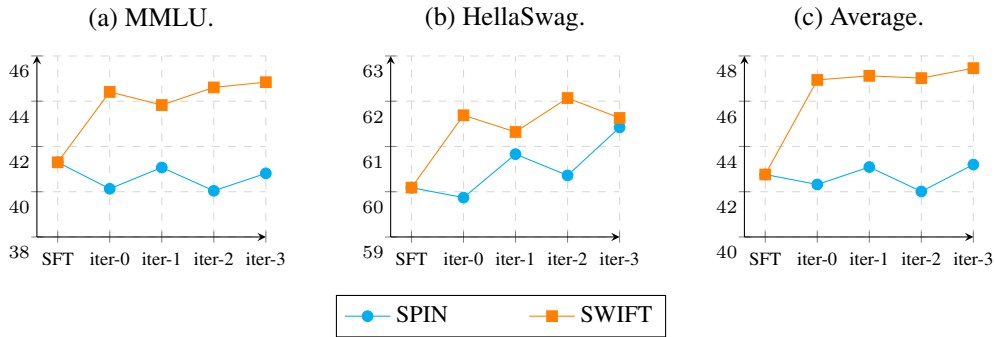

Figure 2: The average score of SPIN and SWIFT at different iterations on the HuggingFace Open LLM leaderboard datasets.

Table 2: Evaluation results of Qwen2.5-7B-Instruct distilled to GPT2-1.5B on four benchmarks and the averaged performance

| | Qwen2.5-7B-Instruct → GPT2-1.5B | | | | |
|---|---|---|---|---|---|
| **Methods** | **Dolly** | **Alpaca** | **S-NI** | **Dialogue Sum** | **Avg** |
| Teacher | $28.49 \pm 0.21$ | $35.75 \pm 0.25$ | $32.35 \pm 0.24$ | $35.24 \pm 0.08$ | 32.96 |
| SFT | $21.83 \pm 0.28$ | $27.15 \pm 0.31$ | $23.16 \pm 0.15$ | $30.74 \pm 0.17$ | 25.72 |
| ULD | $24.52 \pm 0.28$ | $29.17 \pm 0.22$ | $24.18 \pm 0.08$ | $32.74 \pm 0.35$ | 27.65 |
| MinED | $25.52 \pm 0.44$ | $30.41 \pm 0.56$ | $25.09 \pm 0.25$ | $\mathbf{33.83 \pm 0.24}$ | 28.71 |
| DSKD | $25.38 \pm 0.46$ | $30.48 \pm 0.38$ | $25.92 \pm 0.18$ | $33.82 \pm 0.23$ | 28.90 |
| **SWIFT** | $\mathbf{25.94 \pm 0.32}$ | $\mathbf{30.69 \pm 0.33}$ | $\mathbf{26.43 \pm 0.19}$ | $33.74 \pm 0.13$ | **29.20** |

These results highlight an alternative way to leverage teacher models by estimating token-level importance weights, rather than relying solely on logits or hidden states. Moreover, the consistent gains across diverse datasets further demonstrate the robustness of our approach. Furthermore, unlike conventional knowledge distillation, which requires online teacher access and incurs high memory and compute overhead, SWIFT computes token weights in a single offline pass, significantly reducing training costs while retaining the benefits of distillation.

## 4.3 Ablation Studies

We conduct ablation studies to investigate the influence of different token weighting strategies on the performance of SWIFT. Table 3 reports results over four self-play iterations using Qwen1.5-1.8B as the student model under various token importance estimation methods.

We experiment with several token weighting strategies: **random weight**, where weights are uniformly sampled from $[-0.5, 1.5]$; **equal weight**, where all tokens are assigned a constant value of 1; **contrastive weight**, following TIS-DPO [18] by training separate pos-

Table 3: Ablation study for token weight estimation on Qwen1.5-1.8B.

| Methods | iter 0 | iter 1 | iter 2 | iter 3 |
|---|---|---|---|---|
| rand weight | 40.37 | 39.71 | 38.85 | 34.95 |
| equal weight | 42.74 | 43.05 | 43.11 | 43.23 |
| contrastive weight | 44.75 | 45.03 | 45.11 | 45.27 |
| reverse weight | 33.21 | 31.05 | 27.88 | 25.43 |
| **SWIFT** | 46.94 | 47.12 | 47.02 | 47.46 |

itive and negative models to infer token importance; **reverse weight**, which inverts the weights computed by our method. Specifically, if SWIFT assigns a weight $w$ to a token $t$ in sequence $y$, the reverse weighting assigns it $1 - w$; and **SWIFT**, our default setting where token importance is estimated via a teacher model as described in 3.3.

As shown in Table 3, our method (SWIFT) consistently achieves the best performance across all iterations. Equal weighting performs reasonably but is clearly outperformed by importance-aware alternatives. Contrastive weighting provides moderate gains but still trails behind our approach,

while both random and reverse weighting lead to substantial performance degradation. These results underscore the importance of accurate token importance estimation, and confirm the advantage of using a teacher-guided approach as in SWIFT.

We further examine the evolution of chosen and rejected rewards throughout training, as depicted in Figure 1b. Inspire by [18], in our approach, the chosen reward is computed by incorporating token-level weights into the DPO reward formulation, specifically: $\sum_{i=1}^{T^w} w_i^w \beta \log \frac{\pi_\theta^*(y_i^w|x,y_{<i}^w)}{\pi_{\text{ref}}(y_i^w|x,y_{<i}^w)}$. In SPIN, both chosen and rejected rewards decline over time, reflecting ineffective learning of preferred responses. In contrast, when applying our token weighting strategy, the chosen reward increases steadily while the rejected reward decreases, indicating that incorporating token-level importance helps guide the model toward more effective optimization.

## 5 Conclusion

We presented SWIFT, an effective method for token-level importance-aware fine-tuning via teacher-guided distillation. By estimating token weights from a stronger teacher model, SWIFT improves model alignment and generation quality. Our results across multiple benchmarks and settings validate the effectiveness and flexibility of this approach. However, using the teacher model repeatedly across multiple self-play iterations may introduce additional overhead. Future work may explore more efficient strategies, as well as deeper analysis on how token weighting influences learning dynamics.

## Acknowledgements

Trung Le and Mehrtash Harandi were supported by the ARC Discovery Project grants DP230101176 and DP250100262, as well as by the Air Force Office of Scientific Research under award number FA9550-23-S-0001.

We thank the anonymous NeurIPS 2025 reviewers (Reviewer Qm7U, 8XmY, Qi3n, ti82) and Program Chair for their constructive feedback and valuable suggestions that have substantially improved this manuscript.

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

# A  Background

We consider a Large Language Model (LLM) parameterized by parameters $\theta$, and denote its output distribution as $\pi_\theta$. Given an input sequence $x$, commonly known as the prompt, the model generates a response sequence $y$. In preference-based alignment, the training dataset is made up of triplets $\mathcal{D} = \{(x, y_w, y_l)\}$, where $y_w$ and $y_l$ are two possible responses. Among them, $y_w$ is considered better than $y_l$, which we write as $y_w \succ y_l$. We refer to $y_w$ as the chosen (winning) response and $y_l$ as the rejected (losing) one. In the following sections, we provide a brief overview of DPO, SPIN, recent token-level extensions of DPO and traditional knowledge distillation approaches.

**Direct Preference Optimization (DPO).**   DPO [16] provides an elegant and efficient alternative to RLHF [11] by avoiding explicit reward model training. Instead, it reformulates the reward function $r(x, y)$ using a closed-form expression based on the ratio between the policy model and a fixed reference model:

$$r(x, y) = \beta \log \frac{\pi_\theta(y \mid x)}{\pi_{\text{ref}}(y \mid x)} + \beta \log Z(x), \qquad (17)$$

where $\pi_\theta$ is the current policy model, $\pi_{\text{ref}}$ is a static reference policy, and $Z(x)$ is a partition function independent of the policy. By plugging this reward into the Bradley-Terry framework [43], the preference probability between two responses $y_w$ and $y_l$ is modeled as $p(y_w \succ y_l \mid x) = \sigma\left(r(x, y_w) - r(x, y_l)\right)$, where $\sigma(\cdot)$ denotes the sigmoid function. This leads to the DPO loss function, which optimizes the policy directly using preference data:

$$\mathcal{L}_{\text{DPO}}(\pi_\theta; \pi_{\text{ref}}) = -\mathbb{E}_{(x, y_w, y_l) \sim \mathcal{D}}\left[\log \sigma\left(\beta \log \frac{\pi_\theta(y_w \mid x)}{\pi_{\text{ref}}(y_w \mid x)} - \beta \log \frac{\pi_\theta(y_l \mid x)}{\pi_{\text{ref}}(y_l \mid x)}\right)\right], \qquad (18)$$

This formulation enables the model to directly incorporate human preference signals into policy learning without needing an explicit reward model or reinforcement learning.

**Self-Play Fine-Tuning.**   Inspired by Generative Adversarial Networks (GAN) [44], Self-Play fInetuNing (SPIN) [19] proposes an iterative self-play framework where a language model fine-tunes itself by generating synthetic responses and learning to distinguish them from human-annotated data. Specifically, at each iteration $t$, the current model $\pi_{\theta_t}$ generates a response $y'$ for a given prompt $x$, forming a synthetic preference pair $(x, y \succ y')$, where $y$ is the ground-truth response. The model is then updated by minimizing:

$$\mathcal{L}_{\text{SPIN}}(\pi_{\theta_{t+1}}, \pi_{\theta_t}) = \mathbb{E}_{x, y \sim p_{\text{data}}, y' \sim \pi_{\theta_t}}\left[\ell\left(\beta \log \frac{\pi_{\theta_{t+1}}(y \mid x)}{\pi_{\theta_t}(y \mid x)} - \beta \log \frac{\pi_{\theta_{t+1}}(y' \mid x)}{\pi_{\theta_t}(y' \mid x)}\right)\right], \qquad (19)$$

where $\ell(\cdot)$ is a convex decreasing loss (e.g., logistic loss), and $\beta$ is a scaling factor. This formulation encourages the updated model to assign higher likelihood to responses resembling the ground-truth and lower likelihood to its own earlier responses. This self-play process eliminates the need for explicit reward models or preference-labeled data.

**Token-Level DPO.**   Recent works have recognized the value of fine-grained feedback. Rafailov et al. [16] theoretically demonstrate that DPO can represent any dense reward function by reparameterizing it as an optimal advantage function within a token-level Markov Decision Process. This formulation enables DPO to effectively optimize policies at the token level. Building upon this, TDPO [17] introduces forward Kullback-Leibler divergence constraints and leverages the Bradley-Terry model to convert sentence-level preferences into token-level rewards, allowing the model to adjust its strategy dynamically during generation. Furthermore, TIS-DPO [18] enhances this approach by estimating token importance weights based on differences in prediction probabilities from contrastive language models, enabling importance sampling that approximates the optimal distribution by assigning weights to each token according to its estimated reward.

**Traditional Knowledge Distillation Approaches.** Knowledge distillation (KD) is a widely used technique to transfer knowledge from a larger teacher model to a smaller student model. Traditionally, KD employs the Kullback-Leibler (KL) divergence to minimize the difference between the teacher and student probability distributions [21]). Given a sequence $\mathbf{x}$, the student model learns to match the teacher's output distribution by minimizing the following loss: $\mathcal{L}_{\text{KD}} = \sum_i D_{KL}(p(x_i \mid \mathbf{x}_{<i}, \tau) \,\|\, q_\theta(x_i \mid \mathbf{x}_{<i}, \tau))$, where $D_{KL}(\cdot \,\|\, \cdot)$ denotes the KL divergence and $\tau$ is the temperature to control the smoothness of the distributions.

# B   Implementation Details and Algorithm

## B.1   Experiments Setup

We conduct two experimental settings to evaluate SWIFT: **Alignment** and **Knowledge Distillation**. The Alignment setting compares SWIFT with existing alignment methods, while the Knowledge Distillation setting assesses its effectiveness against existing knowledge distillation methods.

For the **Alignment** setting, we use Qwen1.5-1.8B [26] as the base model. As the teacher model, we adopt Zephyr-7B-SFT-Full [27], which is based on Mistral-7B [28] and further fine-tuned on the `Ultrachat200k` dataset[2] provided by HuggingFace. `Ultrachat200k` is a curated 200k subset of the UltraChat corpus [45], which consists of approximately 1.4 million high-quality instructional dialogues generated via OpenAI's Turbo API. We follow the procedure in [19], 50,000 prompts are randomly sampled from `Ultrachat200k` and generate synthetic responses using the base model. For evaluation, we adopt the HuggingFace Open LLM Leaderboard [29], implemented via the Language Model Evaluation Harness [46]. The leaderboard covers six representative benchmarks targeting different capabilities of LLMs: commonsense reasoning (ARC [47], HellaSwag [48], Winogrande [49]), multi-task language understanding (MMLU [50]), resistance to misinformation (TruthfulQA [51]), and mathematical reasoning (GSM8k [52]). These benchmarks collectively provide a rigorous and diverse framework for evaluating both alignment quality and generalization. Additionally, we evaluated the output quality of the LLM using MT-bench [30] with its provided dataset, we use the API of GPT-4 as the judge. The details of the benchmarks are provided in table 4 below.

Table 4: Details of the HuggingFace Open LLM Leaderboard evaluation datasets, including the number of few-shot examples and the evaluation metric for each.

| **Dataset** | Arc | TruthfulQA | Winogrande | GSM8k | HellaSwag | MMLU |
|---|---|---|---|---|---|---|
| # Few-shot | 25 | 0 | 5 | 5 | 10 | 5 |
| Metric | acc_norm | mc2 | acc | acc | acc_norm | acc |

For the **Knowledge Distillation** setting. We use GPT2-1.5B [31] as the base model and Qwen2.5-7B-Instruct [32] as the teacher. Four datasets are selected for evaluation: `DATABRICKSDOLLY-15K` (**Dolly**) [33], `ALPACA` (**Alpaca**) [34], `S-NI` (**S-NI**) [35], and `DIALOGSUM` (**Dialogsum**) [36]. Final performance is reported using the ROUGE-L metric [37] between generated outputs and human-annotated references. In the state-of-the-art method DSKD [42], distillation is typically performed on a single dataset, while evaluation is conducted across multiple datasets spanning different domains or tasks. In contrast, we construct separate training, validation, and testing splits for each domain, allowing for a more targeted evaluation of knowledge distillation within the same domain. The details of the datasets are provided in table 5 below.

Table 5: Dataset Statistics

| **Dataset** | **Train** | **Validation** | **Test** |
|---|---|---|---|
| Dolly | 11,435 | 1,000 | 500 |
| Alpaca | 10,396 | 500 | 500 |
| S-NI | 10,414 | 500 | 1,902 |
| DialogSum | 12,460 | 500 | 1,500 |

---

[2]`https://huggingface.co/datasets/HuggingFaceH4/ultrachat_200k`

## B.2 Hyperparameters

To reduce training costs and memory consumption, we employ DeepSpeed ZeRO-3 [53] and FlashAttention-2 [54] throughout all training iterations. Models are trained using the RMSProp optimizer [55] without weight decay, following standard practice for LLM alignment fine-tuning. We set the global batch size to 2, use bfloat16 precision, and apply a 10% linear warmup at the start of each iteration. The peak learning rate is set to $5 \times 10^{-7}$ for iterations 0 and 1, and $1 \times 10^{-7}$ for iterations 2 and 3 as training approaches convergence. Each iteration is trained for 2 epochs with a maximum sequence length of 2048 tokens. For token importance estimation as defined in equation 14 in main paper, we set $\mu = 1$, with lower and upper clipping bounds $L = -0.5$ and $U = 1.5$, respectively. The hypeparameter $k$ is fixed to 1. All experiments are conducted on $2 \times$ NVIDIA RTX 4090 GPUs.

## B.3 Synthetic Data Generation

We generate synthetic rejected responses using the library vLLM [56] to speed up inference with distributed inference over multiple GPUs. We use a sampling decoding strategy to generate responses, with a temperature of 1.0 and top_p of 1.0. We consider the prompting template `\n\n<Human>:{prompt}\n\n<Assistant>:`.

## B.4 Algorithm

We provided the pseudocode of our proposed **SWIFT (Self-Play Weighted Fine-Tuning)** method in Algorithm 1.

---

**Algorithm 1** Self-Play Weighted Fine-Tuning (SWIFT)

---

**Require:** $\{(x_i, y_i)\}_{i \in [N]}$: SFT dataset, $\pi_{\theta_0}$: LLM with initial parameters $\theta_0$, $\pi_{\theta_T}$: teacher model, $M$: number of iterations.

1: **for** $t = 0, \ldots, M - 1$ **do**
2:      **for** $i = 1, \ldots, N$ **do**
3:          Generate synthetic data $y_i' \sim \pi_{\theta_t}(\cdot | x_i)$.
4:      **end for**
5:      Compute token importance weights $\omega$ using $\pi_{\theta_T}$ and $\pi_{\theta_t}$
6:      $\theta_{t+1} = \arg\min_{\theta \in \Theta} \sum_{i \in [N]} \left[ l \left( u \left( x, y, y', \pi_{\theta^S}, \omega \right) - v \left( x, y, y', \pi_{\theta^S}, \omega \right) \right) \right]$
7: **end for**
8: **return** $\theta_T$.

---

## B.5 Details of Teacher-Guided Token Importance Estimation

In this section, we provide a detailed explanation of the method described in Section 3.3 (Teacher-Guided Token Importance Estimation) of the main paper, addressing how token-level importance weights are distilled from a teacher model and mapped to student tokens, especially in the presence of tokenizer mismatches. We also include the full algorithm, computational analysis, and illustrative examples to enhance clarity.

### B.5.1 Overview of the Method

Our aim is to develop an efficient mechanism to estimate the importance weight of each token. Intuitively, a well-trained teacher—having been trained on more data with higher model capacity—can better signal which tokens carry higher "reward". In this section, we present a principled method to distill token-level reward signals from a teacher model $\theta^T$ and transfer them to a student model $\theta^S$.

Inspired by TIS-DPO [18], we first estimate the raw importance weight for the $t^{\text{th}}$ token in a response $y = (y_1, \ldots, y_T)$ as

$$\omega_t^{\text{raw}} = k \cdot \exp\left( \mu \cdot \text{clamp}\left( \log \frac{\pi_{\theta^T}(y_t \mid x, y^{<t})}{\pi_{\theta^S}(y_t \mid x, y^{<t})}, L, U \right) \right), \tag{20}$$

where $\mathrm{clamp}(a, L, U) = \min\big(\max(a, L), U\big)$ truncates the log-odds to $[L, U]$ to reduce variance and stabilize optimization. Here $k > 0$ and $\mu$ are constant; $L$ and $U$ are lower and upper clipping bounds.

However, this formulation assumes that the teacher and student models share the same tokenizer and vocabulary. In practice, this assumption rarely holds. A sequence $y$ might be tokenized by the teacher model as $[y_1^T, y_2^T, \ldots, y_t^T]$, and by the student as a different sequence $[y_1^S, y_2^S, \ldots, y_s^S]$, making direct token-to-token alignment in (20) infeasible. To address the misalignment issue caused by differing tokenizers, we propose a generalizable token mapping strategy based on shared surface segments.

**Case 1: Shared Tokenizer.** In the scenario where both the teacher and student models utilize an identical tokenizer, the alignment between their respective token sequences is inherently preserved. Specifically, each token $y_t$ generated by the student model corresponds directly to a token produced by the teacher model at the same position, thereby eliminating the need for additional alignment mechanisms. As a result, the distillation weight $\omega_t$ associated with each token position $t$ can be directly adopted from the unprocessed or raw weighting scheme. Formally, we define:

$$\omega_t = \omega_t^{\mathrm{raw}}, \tag{21}$$

where $\omega_t^{\mathrm{raw}}$ represents the original distillation weight derived from the teacher model's outputs as in 20 without any post-processing or re-alignment.

**Case 2: Different Tokenizers.** When tokenizers differ, direct index alignment between tokens becomes unreliable. Rather than aligning tokens individually, we instead leverage a common structural unit shared across both tokenizations. Most modern tokenization algorithms, including BPE [23], SentencePiece [24], and WordPiece [25], segment text into subwords while generally respecting word boundaries. This means that each token strictly belongs to either a single word or the whitespace separating words, making word spans a natural and reliable basis for alignment. Leveraging this property, we segment the original text into coarse-grained units corresponding to **complete words** and **their leading whitespace**, resulting in a sequence of shared components $\{c_1, c_2, \ldots, c_K\}$. This ensures that every token from both teacher and student sequences can be assigned to one of these shared components, which are small enough to preserve local semantics but broad enough to encompass all subword variations.

Suppose student token $y_i^S$ belongs to component $c_h$, and the teacher tokens in that segment are $T(c_h) = \{y_j^T, y_{j+1}^T, \ldots, y_{j+\ell}^T\}$. Assuming all tokens within a shared component contribute equally, we define the student's token importance as the average of the teacher tokens' weights in the corresponding component $c_h$:

$$\omega_i^S = \frac{1}{|T(c_h)|} \sum_{y_r^T \in T(c_h)} k \cdot \exp\left( \mu \cdot \mathrm{clamp}\left( \log \frac{\pi_{\theta^T}(y_r^T \mid x, y_{<r}^T)}{\pi_{\theta^S}(y_i^S \mid x, y_{<i}^S)}, L, U \right) \right). \tag{22}$$

### B.5.2 Pseudo-Code and Computational analysis for Teacher-Guided Token Weight Estimation

The complete process of the Teacher-Guided Token Weight Estimation method is described in Algorithm 2.

Given a text response $y$, we first tokenize it using the teacher tokenizer $\mathrm{tok}_T$ and the student tokenizer $\mathrm{tok}_S$ to obtain token sequences $Y^T$ and $Y^S$, respectively. In addition, the raw text $y$ is segmented into a sequence of shared components $\mathcal{C}$, where each component consists of a full word along with its leading whitespace. This segmentation **has a time complexity of** $\mathcal{O}(n)$, where $n$ is the number of characters in the input $y$.

For each component $c \in \mathcal{C}$, we define $T(c)$ and $S(c)$ as the sets of teacher and student tokens that belong to $c$. We then will estimate the weight for all the student tokens belonging to this component $c$. Since each token belongs to exactly one component, **the algorithm processes each student token only once**.

The core computation involves measuring the discrepancy between the teacher and student log-likelihoods. For each student token $i \in S(c)$, we compute the log-probability under the student model, and for each teacher token $t \in T(c)$, we compute its log-probability under the teacher

---

**Algorithm 2** Teacher-Guided Token Importance Estimation

---

**Require:** Teacher model $\pi_{\theta_T}$, student model $\pi_{\theta_S}$, teacher tokenizer $\text{tok}_T$, student tokenizer $\text{tok}_S$,
hyperparameters $(k, \mu, L, U)$, response $y$.
**Ensure:** Importance weights $\omega^S_{1:|y^S|}$ for student tokens.

1: $Y^T \leftarrow \text{tok}_T(y), \quad Y^S \leftarrow \text{tok}_S(y)$
2: $\mathcal{C} \leftarrow \text{WORDSEGMENTS}(y)$       ▷ Segment raw text into word-level spans
3: **for** each component $c$ in $\mathcal{C}$ **do**
4:    $T(c) \leftarrow \{ t : Y^T_t \subset c \}, \quad S(c) \leftarrow \{ i : Y^S_i \subset c \}$
5:    **for** each $t \in T(c)$ **do**
6:      $a_t \leftarrow \log(\pi_{\theta_T}(Y^T_t \mid x, Y^T_{<t}))$
7:    **end for**
8:    **for** each $i \in S(c)$ **do**
9:      $b_i \leftarrow \log(\pi_{\theta_S}(Y^S_i \mid x, Y^S_{<i}))$
10:      **for** each $t \in T(c)$ **do**
11:        $\ell_{t;i} \leftarrow a_t - b_i$
12:        $\omega^{\text{raw}}_t \leftarrow k \cdot \exp(\mu \cdot \text{clamp}(\ell_{t;i}, L, U))$
13:      **end for**
14:      $\omega^S_i \leftarrow \frac{1}{|T(c)|} \sum_{t \in T(c)} \omega^{\text{raw}}_t$       ▷ eq 22
15:    **end for**
16: **end for**
17: **return** $\{\omega^S_i\}^{|Y^S|}_{i=1}$

---

model. The weight for student token $i$ is then obtained by averaging the clamped exponential of the log-probability differences between the teacher and student tokens within the same component, as formalized in Equation 22.

This design has two critical efficiency advantages: (i) the decoupled log probability computation allows **for only a single forward pass** per model per response, and (ii) the local averaging over shared components yields robust token importance estimates. These properties make the algorithm both scalable and robust to differences in tokenization schemes between teacher and student models.

### B.5.3   Illustrative Examples

Figure 3 illustrates our Teacher-Guided Token Weight Estimation method applied to the input `"Taylor Swift is a singer not an AI algorithm"`. The sentence is segmented into shared components (green), which serve as alignment components between teacher tokens (red) and student tokens (blue). For each component, token-level log-probabilities are computed from both models. Student token weights are then computed using a divergence function $D(x) = k \cdot \exp(\mu \cdot \text{clamp}(x, L, U))$ over the difference between teacher and student log-probabilities. In cases where multiple tokens align to the same component, weights are averaged across the corresponding pairs.

Note that, there may be more than one whitespace character between two words, or including newline characters (e.g., \n). Importantly, modern tokenizers do not tokenize words alone—they also treat sequences of whitespace (spaces, tabs, or newlines) as separate tokens. In our implementation, we preserve these segments as standalone components, denoted as `[SPACE]`.

### B.5.4   Comparison with Prior Work.

Compared to the contrastive-based token weighting in [18], our method is significantly more computationally efficient, as it does not require training separate positive and negative models. We validate this by providing a direct comparison of the total time cost between SWIFT and the contrastive-based token weighting approach (TIS-DPO) in Appendix C.1 . Furthermore, as shown in Table 3 in main paper, our teacher-guided estimation consistently achieves superior performance, confirming both its efficacy and practicality.

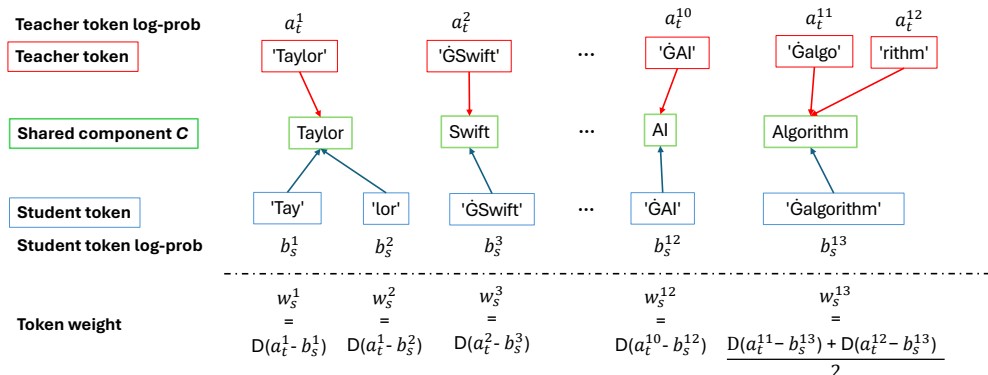

Figure 3: An example of our Teacher-Guided Token Weight Estimation method where $D(x) = k \cdot \exp\big(\mu \cdot \mathrm{clamp}(x, L, U)\big)$

## C Additional Experiment Result

### C.1 Training Overhead

To further examine the computational efficiency of our method, we compare the total runtime of the full training pipeline between SWIFT and the contrastive-based token weighting approach (TIS-DPO), as introduced in [18]. The TIS-DPO framework comprises three strategies for estimating token weights: (1) TIS-DPO (P), which guides the original LLM with contrastive prompts; (2) TIS-DPO (S), which involves training two separate LLMs via SFT on winning and losing responses, respectively; and (3) TIS-DPO (D), which performs both forward and reverse DPO training using winning and losing responses. Among these, [18] report that TIS-DPO (D) achieves the strongest empirical performance, and thus we adopt it for our comparison.

Table 6: Total times for different iterations under SWIFT and TIS-DPO compute weight method.

| Iteration | SWIFT | | | TIS-DPO (D) | | |
|---|---|---|---|---|---|---|
| | Generation | Compute Weight | Training | Generation | Compute Weight | Training |
| Iter 0 | 0.52h | 3.08h | 6.03h | 0.52h | 15.33h | 6.16h |
| Iter 1 | 0.51h | 3.05h | 6.13h | 0.52h | 15.24h | 6.10h |
| Iter 2 | 0.52h | 3.10h | 6.01h | 0.50h | 15.15h | 6.08h |
| Iter 3 | 0.49h | 3.05h | 6.10h | 0.51h | 15.28h | 6.06h |

Table 6 presents the generation, weight computation, and training times for both SWIFT and TIS-DPO (D) across different self-play iterations. Notably, while generation and training times are comparable between the two approaches, SWIFT demonstrates a significant reduction in weight computation time. This efficiency gain arises because TIS-DPO (D) requires training two distinct models to obtain token weights. This highlights the superior computational efficiency of our approach over TIS-DPO (D). Moreover, as shown in Table 3 in main paper, SWIFT not only offers reduced overhead but also achieves better overall performance.

To further quantify the computational overhead of SWIFT, we report the GPU-hours required for each stage of the SWIFT pipeline on the 50k Ultrachat subset using a single NVIDIA H100 GPU. The main phases include: (1) response generation, (2) token weight computation (which includes the teacher forward pass), and (3) training. The breakdown per iteration is shown below:

As shown, the teacher forward pass (offline, single forward) only takes $\tilde{0}.19$ hours $\tilde{1}1$ minutes for 50k samples, which we believe is quite efficient and does not contribute significantly to the overall

Table 7: GPU-hours per iteration for each stage of SWIFT on 50k Ultrachat samples.

| Iteration | Generation (1) | Compute Weight (2) | Training (3) | Teacher Forward | Overall |
|---|---|---|---|---|---|
| 0 | 0.24 h | 0.58 h | 1.45 h | 0.19 h | 2.27 h |
| 1 | 0.23 h | 0.57 h | 1.42 h | 0.19 h | 2.22 h |
| 2 | 0.24 h | 0.58 h | 1.43 h | 0.19 h | 2.25 h |
| 3 | 0.22 h | 0.55 h | 1.40 h | 0.18 h | 2.17 h |

training cost. Notably, in this table, we only used sequential (single-process) inference for simplicity; parallel inference would further reduce this time substantially.

To compare efficiency with distillation baselines, we report the training time per sample and peak GPU memory usage across SFT, ULD, DSKD, and SWIFT. However, we would like to note that our distillation codebase is based on DSKD, which uses a different training library than our SWIFT codebase, making direct comparisons challenging. However, we made every effort to fairly compare the training time per sample and peak GPU memory usage across these baselines in the table below:

Table 8: Training time per sample and peak GPU memory usage.

| Method | Training Time / Sample (s) | Peak GPU Memory (GB) |
|---|---|---|
| SFT | 0.26 | 51.08 |
| ULD | 0.60 | 70.20 |
| DSKD | 0.54 | 66.63 |
| SWIFT | 0.30 | 55.39 |

These results show that SWIFT is comparable to SFT in runtime and memory usage, while more efficient than typical distillation baselines.

## C.2 Robustness to Noisy Guidance

To examine the stability of SWIFT under imperfect guidance, we inject uniform noise in the range $[-0.2, 0.2]$ into the teacher-estimated token weights. As shown in Table 9, performance decreases slightly under noise, but SWIFT remains stable across iterations, indicating resilience to moderate perturbations in teacher signals.

Table 9: Performance comparison under 20% annotation noise

| Noise | Iter | Arc | TruthfulQA | Winogrande | GSM8k | MMLU | HellaSwag | Avg |
|---|---|---|---|---|---|---|---|---|
| 0 | ite0 | 39.16 | 41.01 | 61.96 | 33.43 | 44.41 | 61.69 | 46.94 |
| 0 | ite1 | 38.91 | 38.93 | 61.80 | 37.91 | 43.83 | 61.32 | 47.12 |
| 0 | ite2 | 39.76 | 40.02 | 62.04 | 33.60 | 44.61 | 62.07 | 47.02 |
| 0 | ite3 | 39.78 | 39.12 | 61.48 | 37.93 | 44.84 | 61.63 | 47.46 |
| $\pm0.2$ | ite0 | 37.07 | 38.37 | 62.09 | 32.22 | 44.06 | 60.60 | 45.74 |
| $\pm0.2$ | ite1 | 37.58 | 38.95 | 61.75 | 34.51 | 43.45 | 60.85 | 46.18 |
| $\pm0.2$ | ite2 | 37.63 | 39.54 | 62.13 | 33.19 | 44.17 | 61.48 | 46.36 |
| $\pm0.2$ | ite3 | 38.06 | 39.08 | 62.01 | 35.18 | 44.09 | 61.53 | 46.66 |

Moreover, we would like to clarify why our method might actually have an advantage in lower-data regimes. A crucial premise of this paper is that ground truth can contain some low-reward tokens. These tokens can be understood as noise in SPIN [19]. By introducing token-level importance estimation, SWIFT can effectively identify and filter out such noisy tokens, optimizing primarily the high-quality parts of responses.

## C.3 Extended Evaluation on Reasoning and Agentic Tasks

To further evaluate the applicability of SWIFT beyond preference alignment, we conduct experiments on reasoning-intensive and agentic settings. For reasoning, we use two challenging benchmarks:

**Big-Bench-Hard (BBH)** [57] and **Discrete Reasoning Over Paragraphs (DROP)** [58]. The student model is `Qwen2.5-7B Instruct` and the teacher model is `Qwen3-32B`. Results show that SWIFT consistently improves over baselines on reasoning tasks.

Table 10: Performance on reasoning benchmarks (higher is better).

| Benchmark | Base | DPO | SPIN | SWIFT |
|---|---|---|---|---|
| BBH | 65.07 | 64.23 | 64.85 | **66.01** |
| DROP | 59.28 | 59.10 | 58.62 | **62.03** |

For agentic capabilities, we evaluate on the **ToolBench** [59] benchmark, which measures performance in tool-augmented interaction tasks. SWIFT again demonstrates consistent improvements.

Table 11: ToolBench evaluation on agentic interaction tasks.

| Metric | Base | DPO | SPIN | SWIFT |
|---|---|---|---|---|
| Act.EM $\uparrow$ | 46.69 | 46.12 | 47.01 | **48.36** |
| F1 $\uparrow$ | 40.51 | 40.63 | 40.73 | **42.08** |
| HalluRate $\downarrow$ | 1.60 | 1.44 | 1.42 | **1.40** |
| Rouge-L $\uparrow$ | 4.35 | 4.57 | 4.40 | **4.61** |

These additional results confirm that SWIFT extends effectively to both complex reasoning tasks and real-world agentic interaction scenarios.

## C.4 Evaluation on Larger Backbone Models

To assess the scalability of SWIFT beyond `Qwen1.5-1.8B` we conduct experiments on recent, stronger instruction-tuned and reasoning-capable backbones. We evaluate two student-teacher configurations: `Qwen3-32B` $\rightarrow$ `Mistral-7B-v0.1` and `Qwen3-32B` $\rightarrow$ `Qwen2.5-7B-Instruct`. Results in table 12 below confirm that SWIFT continues to deliver meaningful performance gains even with newer and stronger instruction and reasoning models.

Table 12: Scaling analysis with larger backbone models.

| Setting | Model | Arc | Truthful | Wino | GSM8k | MMLU | HellaSwag | Avg |
|---|---|---|---|---|---|---|---|---|
| Qwen3-32B | | 71.08 | 59.29 | 76.95 | 80.97 | 81.53 | 83.96 | 75.63 |
| | Base | 59.90 | 42.65 | 77.43 | 31.99 | 60.60 | 83.47 | 59.34 |
| | DPO | 60.70 | 42.11 | 76.59 | 32.65 | 61.30 | **83.92** | 59.55 |
| $\rightarrow$ Mistral-7B-v0.1 | SPIN | 61.73 | 44.63 | 77.07 | 33.14 | 62.76 | 81.68 | 60.17 |
| | SWIFT | **63.06** | **51.25** | **78.20** | **42.20** | **65.17** | 81.13 | **63.50** |
| | Base | 65.96 | 64.70 | 75.06 | 68.61 | **73.49** | 81.45 | 71.55 |
| | DPO | 66.12 | 63.50 | 74.19 | 67.49 | 73.02 | 81.67 | 71.00 |
| $\rightarrow$ Qwen2.5-7B-Instruct | SPIN | 66.53 | 65.76 | 75.17 | 69.35 | 72.38 | 82.30 | 71.92 |
| | SWIFT | **66.98** | **66.21** | **75.53** | **73.95** | 73.36 | **82.45** | **73.08** |

## C.5 Teacher Quality Dependence

To assess the sensitivity of SWIFT to teacher quality, we experiment with multiple teacher models of varying strength and report the resulting student performance after training with each teacher model.

These results show that while stronger teachers yield slightly better student performance, the gains are not strongly correlated with teacher capability. This is because the teacher is only used to estimate the importance weight of token, not to directly control what the model generates. As a result, SWIFT maintains stable and effective performance across diverse teacher–student pairs and does not depend heavily on highly capable teachers.

Table 13: Effect of different teacher models on student performance.

| Setting | Arc | Truthful | Wino | GSM8k | MMLU | HellaSwag | Avg |
|---|---|---|---|---|---|---|---|
| Zephyr-7B-SFT-Full | 60.41 | 43.73 | 74.19 | 26.76 | 60.92 | 82.85 | 58.14 |
| → Qwen1.5-1.8B | 39.78 | 39.12 | 61.48 | 37.93 | 44.84 | 61.63 | 47.46 |
| Qwen2.5-7B-Instruct | 65.96 | 64.70 | 75.06 | 68.61 | 73.49 | 81.45 | 71.54 |
| → Qwen1.5-1.8B | 39.91 | 40.11 | 62.43 | 37.56 | 44.50 | 62.43 | 47.82 |
| Qwen3-32B | 71.08 | 59.29 | 76.95 | 80.97 | 81.53 | 83.96 | 75.63 |
| → Qwen1.5-1.8B | 40.27 | 38.96 | 63.51 | 38.83 | 44.69 | 62.19 | 48.07 |

## C.6  Qualitative Analysis on MMLU benchmarks

To further analyze how SWIFT impacts different knowledge domains, we conducted a qualitative analysis using the MMLU benchmark, which covers diverse knowledge domains. MMLU comprises four major sub-domains: Humanities, Social Sciences, STEM, and Other. Results across iterations are reported in Table 14 below.

Table 14: MMLU subdomain analysis across SWIFT iterations.

| Domain | Base | Ite 0 | Ite 1 | Ite 2 | Ite 3 |
|---|---|---|---|---|---|
| Humanities | 39.85 | **40.89** | 40.13 | 40.70 | 40.78 |
| SocialSci. | 47.43 | 48.29 | 47.90 | 48.88 | **49.59** |
| STEM | 34.61 | 36.76 | 37.08 | 37.93 | **38.12** |
| Other | 50.08 | 52.08 | 52.11 | 51.75 | **52.14** |

The results show clear and consistent improvements, especially in STEM and Social Sciences, which suggests the model is getting better at reasoning and logical thinking. Meanwhile, performance in Humanities and Other domains remains stable or improved, showing that the model's reasoning gains didn't hurt its general knowledge or overall quality.

## C.7  Qualitative Comparison Between Teacher-Guided and Contrastive Token Weighting

To provide further insight into how SWIFT differs from contrastive token weighting (TIS-DPO [18]), we conduct a qualitative analysis comparing the most influential tokens selected by each method.

Specifically, at each iteration, the contrastive weighting strategy adopted from TIS-DPO relies on two separate student models (a positive and a negative model), which are trained via DPO on pairs constructed from (ground_truth, model_response) and (model_response, ground_truth). Token importance is then estimated based on the difference in logits between these two models. While effective initially, this approach faces an inherent limitation as iterations progress: as the model improves over time, its generated responses become increasingly similar to ground truth. Consequently, the distinction between the ground truth and model response training pairs diminishes in later iterations, weakening the contrastive signal and thereby potentially reducing the quality of token importance estimates. In contrast, our proposed SWIFT method employs a fixed, external teacher model and it is unaffected by the student's iterative improvement, it provides stable, high-quality importance signals across iterations.

Furthermore, we performed an additional qualitative analysis, comparing the top 20 highest-weighted tokens from each method. Token rankings were computed based on the average importance per appearance, defined by:

$$\text{score}(t) = \frac{\sum \text{importance weights assigned to } t}{\text{number of appearances of } t}. \tag{23}$$

The results are presented below:

**Top 20 Tokens of SWIFT:**

```
["Serve", "tering", "antibiotic", "intermittent", "evaluates", "destroys",
 "Wealth", "FDA", "angelo", "onne", "visited", "paralyzed", "aken",
 "toughest", "Pricing", "CLA", ".dequeue", "avenous", "plings", "specifies"]
```

**Top 20 Tokens of TIS-DPO:**

```
["FACE", "[@", "LANG", ".dest", "ReturnValue", "baseUrl", "êPrint",
 "keycode", "-has", "HG", "repaint", "Denver", "FINAL", "simil", "ï½s",
 "{/*", "owler", ".des", "Teens", "ÃǦ"]
```

SWIFT selects semantically meaningful and contextually relevant tokens (e.g., `antibiotic`, `FDA`), reflecting focus on factual and domain-rich content. In contrast, TIS-DPO often prioritizes tokens resembling code artifacts or noise (e.g., `ReturnValue`, `@`), suggesting less semantic alignment.

However, we understand that our subjective judgment alone might not be sufficiently convincing, we further conducted an objective evaluation using GPT-4o as an external evaluator to assess each method's top-20 tokens. We designed the following targeted prompt for GPT-4o:

> **Prompt**
>
> Below, I will provide the top 20 most important tokens across an entire dataset, as identified by two different methods. Please evaluate which method is better based on key criteria for token importance in a sentence, such as: semantic relevance (how meaningful or content-rich the token is), syntactic role (its grammatical contribution), contextual influence (how much it affects the surrounding content), and task-specific utility (its contribution to downstream tasks like classification or retrieval). Here are these tokens:
> - top_20_tokens_of_method1: {top_20_tokens_of_SWIFT}
> - top_20_tokens_of_method2: {top_20_tokens_of_TIS-DPO}

GPT-4o provided the following summarized judgment:

> **LLM response**
>
> Method 1 is clearly better overall than Method 2. It includes more semantically rich, syntactically functional, and contextually influential tokens, many of which are likely to be task-relevant (e.g., medical, procedural, or evaluative terms). Method 2 appears to highlight code artifacts, UI labels, or tokenization noise, which are often less useful for language tasks.

### C.8 Generation Examples

Tables 15, 16, and 17 present additional generation examples produced by our SWIFT fine-tuned model across different training iterations. Table 15 showcases examples from math problems, Table 16 focuses on code-generation tasks, and Table 17 includes general instruction-following cases. These examples illustrate a clear improvement in response quality compared to the outputs from the initial SFT checkpoint. Notably, as the number of iterations increases, the model's responses become more concise and better aligned with the ground-truth completions. For clarity, in each prompt, the responses generated by SFT, iteration 0, iteration 1, and iteration 2 are respectively labeled as **Response 1**, **Response 2**, **Response 3**, and **Response 4**.

Table 15: Generation example of our fine-tuned model by SWIFT at different iterations. GPT-4 chooses SWIFT iteration 2 over SFT, and iteration 0, 1.

| | |
|---|---|
| **Prompt** | <Human>:If the year is divisible by both 4 and 100, it is not a leap year, unless it is also divisible by 400.
<Assistant>: |
| **Ground truth** | This is a true statement. For example, the year 1900 was divisible by both 4 and 100 but not by 400, so it was not a leap year. However, the year 2000 was divisible by both 4 and 100 and also by 400, so it was a leap year. |
| **SFT** | This is true. For example, the year 1900 was divisible by both 4 and 100 but not by 400, so it was not a leap year. However, 1904 was divisible by 4, but not by 100, so it was a leap year. The same applies to the year 2000, 2004, and 2008. |
| **Iteration 0** | True. For example, the year 1900 was divisible by both 4 and 100 but not by 400, so it was not a leap year. However, the year 2000 was divisible by both 4 and 100 and also by 400, so it was a leap year. |
| **Iteration 1** | This is a true statement. For example, the year 1900 was divisible by both 4 and 100 but not by 400, so it was not a leap year. However, the year 2000 was divisible by both 4 and 100 and also by 400, so it was a leap year. |
| **Iteration 2** | This is a true statement. For example, the year 1900 was divisible by both 4 and 100 but not by 400, so it was not a leap year. However, the year 2000 was divisible by both 4 and 100 as well as 400, so it was a leap year. |
| **Judgment** | Comparison: Response 3 = Response 4 > Response 2 > Response 1.
Explanation: Responses 3 and 4 are identical and fully match the ground truth both in correctness and clarity, providing the textbook explanation and appropriate examples (1900 and 2000) without unnecessary additions. Response 2 is also correct and concise but slightly less formal in tone by starting with "True" instead of a complete sentence, which slightly reduces its explanatory quality. Response 1, while factually correct in its individual statements, introduces unnecessary examples (1904, 2004, 2008) that deviate from the prompt's focus on the specific exception rule involving divisibility by 100 and 400, potentially confusing the reader by adding standard leap year cases that do not illustrate the special condition described in the prompt. |

Table 16: Generation example of our fine-tuned model by SWIFT at different iterations. GPT-4 chooses SWIFT iteration 2 over SFT, and iteration 0, 1.

| Prompt | <Human>:Write a Python program that prompts the user to enter a string and then counts the number of characters in the entered string. The program should print the total number of characters in the string.
<Assistant>: |
|---|---|
| **Ground truth** | String = input("Enter a string: ")
count = 0
for character in string:
   count += 1
print("The total number of characters in the string is:", count) |
| **SFT** | Here's a possible solution:
string = input("Enter a string: ")
count = 0
for character in string:
   count += 1
print("The total number of characters in the string is:", count)
Here's how the program works:
1. We prompt the user to enter a string using the input() function and store the string in the variable string.
2. We initialize a variable count to 0, which will be used to keep track of the number of characters in the string.
3. We use a for loop to iterate over each character in the string. The loop iterates over each character in the string using string["], which gives us a slice of the string that starts at the current position and goes up to (but not including) the end of the string. We add 1 to the count variable to increment it for each character we encounter.
4. We use the print() function to output the result to the screen. We use string interpolation to embed the count variable in the output string, so it's easy to see how many characters are in the string.
5. The program terminates when the user enters a string, so we use a while loop to wait for user input before printing the final result. |
| **Iteration 0** | String = input("Enter a string: ")
count = 0
for character in string:
   count += 1
print("Total number of characters in the entered string is: ", count) |
| **Iteration 1** | String = input("Enter a string: ")
count = 0
for char in string:
   count += 1
print("The total number of characters in the string is:", count) |
| **Iteration 2** | String = input("Enter a string: ")
count = 0
for character in string:
   count += 1
print("The total number of characters in the string is:", count) |
| **Judgment** | Comparison: Response 4 = Response 3 > Response 2 > Response 1.
Explanation: Responses 4 and 3 are functionally identical to the ground truth, only differing slightly in variable naming ("char" vs. "character") but maintaining correct, concise, and syntactically sound code. Response 2 is also correct but has a minor stylistic difference in the output string ("Total number of characters in the entered string is") that deviates from the ground truth and slightly affects alignment. Response 1, while technically correct in the code block, contains multiple issues in the explanation: it introduces confusion by mentioning a nonexistent while loop and incorrectly describes the use of string["], which is invalid syntax and misrepresents how the loop operates. These inaccuracies make it the least reliable response. |

Table 17: Generation example of our fine-tuned model by SWIFT at different iterations. GPT-4 chooses SWIFT iteration 2 over SFT, and iteration 0, 1.

| | |
|---|---|
| **Prompt** | <Human>:Given the text: During The Daily Show's "Between the Scenes" segment on April 10, Noah asked television mogul Winfrey if there was any sense of normalcy left in her life. The answer, quite simply, was no. Even when she tried to dig up a passable answer, she hilariously failed. The avocado orchard is tucked away on the property of Winfrey's 23,000-square-foot Montecito home, which she purchased for $50 million back in 2001. The home also boasts a 5,000-square-foot ranch-style house, a pool, horse stables, an equestrian ring, a fruit orchard, and more. Simply put, Oprah's home — much like guacamole — is extra. Get it? Could you provide more information about the features of Oprah Winfrey's Montecito property, including the size of her home and other amenities?
<Assistant>: |
| **Ground truth** | Yes, Oprah Winfrey's Montecito property is 23,000-square-foot in size and includes a 5,000-square-foot ranch-style house, a pool, horse stables, an equestrian ring, a fruit orchard, and an avocado orchard. The property was purchased for $50 million in 2001. |
| **SFT** | Yes, the text mentions that Oprah Winfrey purchased her 23,000-square-foot Montecito home for $50 million in 2001. It also states that the home has a 5,000-square-foot ranch-style house, a pool, horse stables, an equestrian ring, a fruit orchard, and an avocado orchard. Therefore, the size of Oprah's home is not given in the text. |
| **Iteration 0** | According to the text, Oprah Winfrey purchased her Montecito property for $50 million in 2001. The property consists of a 23,000-square-foot Montecito home and a 5,000-square-foot ranch-style house, an irrigation pond, natural stone walkways, a fountain, tailgating capability, and more. Additionally, the property includes a fruit orchard and an avocado orchard. |
| **Iteration 1** | Of course! Oprah Winfrey's Montecito property is 23,000-square-foot in size and was purchased for $50 million in 2001. The property also includes a 5,000-square-foot Ranch-style house, a pool, horse stables, an equestrian ring, a fruit orchard, and an avocado orchard. It is truly a sprawling estate that puts even avocado-curious minds to shame. |
| **Iteration 2** | Yes, Oprah Winfrey's Montecito property is 23,000-square-foot in size and includes a 5,000-square-foot ranch-style house, pool, horse stables, an equestrian ring, a fruit orchard, and an avocado orchard. She purchased the property for $50 million in 2001. |
| **Judgment** | Comparison: Response 4 > Response 3 > Response 2 > Response 1.
Explanation: Response 4 is the most accurate, concise, and directly mirrors the ground truth in both factual details and structure, making it the clearest and most reliable answer. Response 3 is similarly factually correct and well-structured but adds unnecessary embellishment ("puts even avocado-curious minds to shame") that, while engaging, introduces subjective commentary not present in the ground truth. Response 2 includes factual inaccuracies (mentions "irrigation pond, natural stone walkways, fountain, tailgating capability") that are not supported by the text, thereby reducing its reliability. Response 1 contains a clear contradiction—despite correctly listing most amenities, it confusingly states "the size of Oprah's home is not given in the text," which is factually incorrect, making it the weakest response. |

# D All Proof

**Proof of Lemma 3.1**

We prove by induction. First, for $t = 1$, we have

$$p_d\left(y_1 \mid x\right) = \frac{p_d\left(x, y_1\right)}{p_d\left(x\right)} \text{ and } p_{\theta^S}\left(y_1 \mid x\right) = \frac{p_{\theta^S}\left(x, y_1\right)}{p_d\left(x\right)}. \tag{24}$$

From Eq. (24), we reach the conclusion for $t = 1$. Assume that we have the conclusion for $i \leq t - 1$. We further derive

$$p_d\left(x, y_{\leq t}\right) = p_d\left(y_t \mid x, y_{<t}\right) p_d\left(x\right) \prod_{i=1}^{t-1} p_d\left(y_i \mid x, y_{<i}\right),$$

$$p_{\theta^S}\left(x, y_{\leq t}\right) = p_{\theta^S}\left(y_t \mid x, y_{<t}\right) p_d\left(x\right) \prod_{i=1}^{t-1} p_{\theta^S}\left(y_i \mid x, y_{<i}\right). \tag{25}$$

From Eq. (25), we reach the conclusion for $t$.

**Proof of Theorem 3.3**

Let $r_t^* \in \mathcal{R}_t^k$ be the solution of

$$\forall t : \max_{r_t \in \mathcal{R}_t^k} \mathbb{E}_{[x, y_{\leq t}] \sim D} \left[ r_t\left([x, y_{<t}], y_t\right) - \mathbb{E}_{y_t' \sim \pi_{\theta_k^S}(\cdot \mid x, y_{<t})} \left[r_t\left[x, y_{<t}\right], y_t'\right] \right]. \tag{26}$$

According to the consistency property, there exists $r^* \in \mathcal{R}^k$ such that $\forall t : r_t^*\left([x, y_{<t}], y_t\right) = r^*\left(x, y_{\leq t}\right)$. We now prove that $r^*$ is the solution of the following OP:

$$\max_{r \in \mathcal{R}^k} \mathbb{E}_{(x,y) \sim D, y' \sim \pi_{\theta_k^S}(\cdot \mid x)} \left[ \sum_t \gamma^{t-1} r\left([x, y_{<t}], y_t\right) - \sum_t \gamma^{t-1} r\left([x, y_{<t}], y_t'\right) \right], \tag{27}$$

where we denote $r\left([x, y_{<t}], y_t\right) = r\left(x, y_{\leq t}\right)$.

Let $\bar{r}^* \in \mathcal{R}^k$ be the solution of the OP in (27). We have

$$\sum_t \gamma^{t-1} \bar{r}^*\left([x, y_{<t}], y_t\right) - \sum_t \gamma^{t-1} \bar{r}^*\left([x, y_{<t}], y_t'\right) = \sum_t \gamma^{t-1} \left[\bar{r}^*\left([x, y_{<t}], y_t\right) - \bar{r}^*\left([x, y_{<t}], y_t'\right)\right]$$

$$= \sum_t \gamma^{t-1} \left[\bar{r}_t^*\left([x, y_{<t}], y_t\right) - \bar{r}_t^*\left([x, y_{<t}], y_t'\right)\right] \leq \sum_t \gamma^{t-1} \left[r_t^*\left([x, y_{<t}], y_t\right) - r_t^*\left([x, y_{<t}], y_t'\right)\right]$$

$$= \sum_t \gamma^{t-1} \left[r^*\left([x, y_{<t}], y_t\right) - r^*\left([x, y_{<t}], y_t'\right)\right], \tag{28}$$

where $\bar{r}_t^* \in \mathcal{R}_t^k$ are the consistent versions of $\bar{r}^* \in \mathcal{R}^k$ in the token-level families.

Finally, (28) indicates that $r^*$ is the solution of the OP in (27).

**Proof of Theorem 3.4**

We consider

$$d_1\left(\mathbb{P}_d, \mathbb{P}_{\theta^S}\right) := \max_{r \in \mathcal{R}^k} \mathbb{E}_{(x,y) \sim D, y' \sim \pi_{\theta_k^S}(\cdot \mid x)} \left[R\left(x, y\right) - R(x, y')\right]. \tag{29}$$

Because if $r \in \mathcal{R}^k$ then $-r \in \mathcal{R}^k$, we have $d_1\left(\mathbb{P}_d, \mathbb{P}_{\theta^S}\right) \geq 0$ and $\min_{\theta^S \in \Theta} d_1\left(\mathbb{P}_d, \mathbb{P}_{\theta^S}\right) \geq 0$. Moreover, the minimization is obtained at 0 for $\bar{\theta}^S$ such that $\mathbb{P}_{\bar{\theta}^S} = \mathbb{P}_d$.

We now consider

$$d_2\left(\mathbb{P}_d, \mathbb{P}_{\theta^S}\right) := \max_{r \in \mathcal{R}^k} \mathbb{E}_{(x,y) \sim D, y' \sim \pi_{\theta_k^S}(\cdot \mid x)} \left[ \sum_t \gamma^{t-1} r\left([x, y_{<t}], y_t\right) - \sum_t \gamma^{t-1} r\left([x, y_{<t}], y_t'\right) \right], \tag{30}$$

Because if $r \in \mathcal{R}^k$ then $-r \in \mathcal{R}^k$, we have $d_2\left(\mathbb{P}_d, \mathbb{P}_{\theta^S}\right) \geq 0$ and $\min_{\theta^S \in \Theta} d_2\left(\mathbb{P}_d, \mathbb{P}_{\theta^S}\right) \geq 0$. Moreover, the minimization is obtained at 0 for $\bar{\theta}^S$ such that $\mathbb{P}_{\bar{\theta}^S} = \mathbb{P}_d$ according to Lemma 1.

**Proof of Lemma 3.5**

Our proof is adopted from the proof of Lemma 4.2 in [17].

$$\max_{\pi_{\theta^S}} \mathbb{E}_{x,y_{<t}\sim D, y_t'\sim\pi_{\theta^S}(\cdot|x,y_{<t})} \left[ \omega_t A_{\pi_{\theta_k^S}}\left([x,y_{<t}],y_t'\right) \right] - \beta D_{KL}\left(\pi_{\theta^S}\left(\cdot\mid[x,y_{<t}]\right)\|\pi_{\theta_k^S}\left(\cdot\mid[x,y_{<t}]\right)\right)$$

$$= \max_{\pi_{\theta^S}} \mathbb{E}_{x,y_{<t}\sim D, y_t'\sim\pi_{\theta^S}(\cdot|x,y_{<t})} \left[ \omega_t Q_{\pi_{\theta_k^S}}\left([x,y_{<t}],y_t'\right) - \omega_t V_{\pi_{\theta_k^S}}\left([x,y_{<t}]\right) - \beta\log\frac{\pi_{\theta^S}\left(y_t'\mid[x,y_{<t}]\right)}{\pi_{\theta_k^S}\left(y_t'\mid[x,y_{<t}]\right)} \right]$$

$$= \max_{\pi_{\theta^S}} \mathbb{E}_{x,y_{<t}\sim D, y_t'\sim\pi_{\theta^S}(\cdot|x,y_{<t})} \left[ \beta\log\frac{\pi_{\theta_k^S}\left(y_t'\mid[x,y_{<t}]\right)\exp\left\{\frac{\omega_t}{\beta}Q_{\pi_{\theta_k^S}}\left([x,y_{<t}],y_t'\right)\right\}}{\pi_{\theta^S}\left(y_t'\mid[x,y_{<t}]\right)Z\left([x,y_{<t}];\omega_t,\beta\right)} \right.$$
$$\left. -\omega_t V_{\pi_{\theta_k^S}}\left([x,y_{<t}]\right) + \log Z\left([x,y_{<t}];\omega_t,\beta\right) \right]$$

$$= \max_{\pi_{\theta^S}} \mathbb{E}_{x,y_{<t}\sim D, y_t'\sim\pi_{\theta^S}(\cdot|x,y_{<t})} \left[ -\beta D_{KL}\left(\pi_{\theta^S}\left(y_t'\mid[x,y_{<t}]\right)\|\frac{\pi_{\theta_k^S}\left(y_t'\mid[x,y_{<t}]\right)\exp\left\{\frac{\omega_t}{\beta}Q_{\pi_{\theta_k^S}}\left([x,y_{<t}],y_t'\right)\right\}}{Z\left([x,y_{<t}];\omega_t,\beta\right)}\right) \right.$$
$$\left. -\omega_t V_{\pi_{\theta_k^S}}\left([x,y_{<t}]\right) + \log Z\left([x,y_{<t}];\omega_t,\beta\right) \right].$$

This concludes our proof.

**Proof of Lemma 3.6**

Using the same derivations as in [17], we gain

$$R(x,y) = V_\pi\left([x,y^{<1}]\right) - \gamma^T V_\pi\left([x,y_{<T+1}]\right)$$
$$+ \sum_{t=1}^{T}\left[\gamma^{t-1}\left(r\left([x,y_{<t}],y_t\right)+\gamma V_\pi\left([x,y_{<t+1}]\right)\right)-V_\pi\left([x,y_{<t}]\right)\right]$$
$$= V_\pi\left([x,y^{<1}]\right) - V_\pi\left([x,y_{<T+1}]\right)$$
$$+ \sum_{t=1}^{T}\left[r\left([x,y_{<t}],y_t\right)+V_\pi\left([x,y_{<t+1}]\right)-V_\pi\left([x,y_{<t}]\right)\right],$$

since $\gamma=1$.

We further have

$$Q_\pi\left([x,y_{<t}],y_t\right) = r\left([x,y_{<t}],y_t\right)+V_\pi\left([x,y_{<t+1}]\right),$$
$$A_\pi\left([x,y_{<t}],y_t\right) = Q_\pi\left([x,y_{<t}],y_t\right)-V_\pi\left([x,y_{<t}]\right)$$
$$= r\left([x,y_{<t}],y_t\right)+V_\pi\left([x,y_{<t+1}]\right)-V_\pi\left([x,y_{<t}]\right).$$

By noting that

$$V_\pi\left([x,y_{<T+1}]\right) = 0 \text{ and } V_\pi\left([x,y^{<1}]\right)=V_\pi\left([x,y'^{<1}]\right)=V_\pi\left([x,[]]\right),$$

we gain

$$l\left(R(x,y)-R(x,y')\right) = l\left(\sum_t A_\pi\left([x,y_{<t}],y_t\right)-\sum_t A_\pi\left([x,y'_{<t}],y_t'\right)\right).$$

**Proof of Theorem 3.7**

We have

$$Q_{\pi_{\theta_k^S}}\left([x,y_{<t}],z\right) = \omega_t^{-1}\beta\log\frac{\pi_{\theta^S}^*\left(z\mid x,y_{<t}\right)}{\pi_{\theta_k^S}\left(z\mid x,y_{<t}\right)}+\omega_t^{-1}\beta\log Z\left([x,y_{<t}];\omega_t,\beta\right). \qquad (31)$$

We also have

$$l\left(R(x,y)-R(x,y')\right) = l\left(\sum_t\gamma^{t-1}A_{\pi_{\theta_k^S}}\left([x,y_{<t}],y_t\right)-\sum_t\gamma^{t-1}A_{\pi_{\theta_k^S}}\left([x,y'_{<t}],y_t'\right)\right). \qquad (32)$$

We further derive as

$$\sum_t \gamma^{t-1} A_{\pi_{\theta_k^S}}\left([x, y_{<t}], y_t\right) = \sum_t \gamma^{t-1}\left[Q_{\pi_{\theta_k^S}}\left([x, y_{<t}], y_t\right) - V_{\pi_{\theta_k^S}}\left([x, y_{<t}]\right)\right]$$

$$= \sum_t \gamma^{t-1}\left[Q_{\pi_{\theta_k^S}}\left([x, y_{<t}], y_t\right) - \mathbb{E}_{y_t' \sim \pi_{\theta_k^S}}\left[Q_{\pi_{\theta_k^S}}\left([x, y_{<t}], y_t'\right)\right]\right]$$

$$= \sum_t \gamma^{t-1}\left[\omega_t^{-1}\beta \log \frac{\pi_{\theta^S}^*\left(y_t \mid x, y_{<t}\right)}{\pi_{\theta_k^S}\left(y_t \mid x, y_{<t}\right)} + \omega_t^{-1}\beta \log Z\left([x, y_{<t}]; \omega_t, \beta\right)\right.$$

$$\mathbb{E}_{y_t' \sim \pi_{\theta_k^S}}\left[\omega_t^{-1}\beta \log \frac{\pi_{\theta^S}^*\left(y_t' \mid x, y_{<t}\right)}{\pi_{\theta_k^S}\left(y_t' \mid x, y_{<t}\right)} + \omega_t^{-1}\beta \log Z\left([x, y_{<t}]; \omega_t, \beta\right)\right]\right]$$

$$= \sum_t \gamma^{t-1}\left[\omega_t^{-1}\beta \log \frac{\pi_{\theta^S}^*\left(y_t \mid x, y_{<t}\right)}{\pi_{\theta_k^S}\left(y_t \mid x, y_{<t}\right)} - \mathbb{E}_{y_t' \sim \pi_{\theta_k^S}}\left[\omega_t^{-1}\beta \log \frac{\pi_{\theta^S}^*\left(y_t' \mid x, y_{<t}\right)}{\pi_{\theta_k^S}\left(y_t' \mid x, y_{<t}\right)}\right]\right]$$

$$= \sum_t \gamma^{t-1}\left[\omega_t^{-1}\beta \log \frac{\pi_{\theta^S}^*\left(y_t \mid x, y_{<t}\right)}{\pi_{\theta_k^S}\left(y_t \mid x, y_{<t}\right)} + \omega_t^{-1}\beta D_{KL}\left(\pi_{\theta^S}^*\left(\cdot \mid x, y_{<t}\right) \| \pi_{\theta_k^S}\left(\cdot \mid x, y_{<t}\right)\right)\right]$$

$$= \beta \sum_t \gamma^{t-1}\omega_t^{-1} \log \frac{\pi_{\theta^S}^*\left(y_t \mid x, y_{<t}\right)}{\pi_{\theta_k^S}\left(y_t \mid x, y_{<t}\right)} + \beta \sum_t \omega_t^{-1}\gamma^{t-1} D_{KL}\left(\pi_{\theta^S}^*\left(\cdot \mid x, y_{<t}\right) \| \pi_{\theta_k^S}\left(\cdot \mid x, y_{<t}\right)\right).$$

Finally, substituting to Eq. (32), we gain the conclusion.

