# OpenReview forum: "Token-Level Self-Play with Importance-Aware Guidance for Large Language Models"
_NeurIPS.cc/2025/Conference — NeurIPS 2025 poster_

### Official Review · Reviewer_ti82 · 2025-06-25

**Clarity:** 2
**Significance:** 3
**Originality:** 3
**Rating:** 4
**Confidence:** 3

**Summary:**

To address the issues of the Self-Play Fine-Tuning, authors want to put different weight into each token. This paper proposes self-play weighted fine-tuning to incorporate token-level importance weights from a stronger teacher model. Considering the mismatches of tokenizer between the teacher model and the student model, authors introduce a practical solution in two different cases. Comprehensive experiments shows improvement compared to DPO-related alignment methods.

**Questions:**

1. In the proof of Lemma 3.1 in the Appendix, could you provide further clarification on why the existing proof is sufficient to establish the necessary and sufficient condition? Specifically, what steps or assumptions are being made that allow for this conclusion?
2. Could you elaborate on the relationship between the proposed approach and the theorems presented? A more explicit expression or explanation would help to clarify how these components interact and support the overall method.
3. While the experiments demonstrate improvements on small language models, it would be valuable to investigate the performance of the proposed approach on larger models. How does the method scale, and are there any limitations or challenges that arise when applying it to more complex models?
4. What is the computational efficiency of the proposed method compared to existing baselines? Are there any trade-offs between accuracy and computational resources, and if so, how do they impact the practicality of the approach?

**Ethical Concerns:**

["NO or VERY MINOR ethics concerns only"]

**Final Justification:**

In the rebuttal phase, new experiments addressed the limitation of the initial paper, supporting their claim on effectiveness. I suggest accepting the paper.

**Quality:**

3

**Strengths And Weaknesses:**

Pros:
1. The motivation behind assigning different weights to individual tokens is straightforward and intuitive.
2. In addition to the proposed approaches, the authors provide theoretical guidance and consider real-world challenges, such as mismatches between tokenizers. They also present ablation experiments that exclude distillation from the teacher models, offering further insights into their method.
3. The baselines used in the study are comprehensive, allowing for a thorough validation of the model's performance.

Cons:
1. The writing is unclear in certain sections, leading to confusion and making it difficult to follow the authors' arguments.
2. A limitation of the experimental setup is that the student model used is relatively small, with approximately 1 billion parameters, which may not accurately represent more complex models.
3. The proposed method heavily relies on previous work (self-play fine-tuning), which somewhat limits its technical contribution and novelty.

---

> ### Author Rebuttal · Authors · 2025-07-31
>
> Dear reviewer **ti82**, thank you for your thoughtful review and for recognizing our work. Below, we address your questions and concerns in detail.
>
> ---
> **Response to Weakness1.**
>
> > 1. The writing is unclear in certain sections, leading to confusion and making it difficult to follow the authors' arguments.
>
> **A1:** Thank you for the feedback. We will carefully revise all the sections to clarify our arguments and improve overall readability.
>
> ---
> **Response to Weakness2.**
>
> > 2. A limitation of the experimental setup is that the student model used is relatively small, with approximately 1 billion parameters, which may not accurately represent more complex models.
>
> **A2:** We agree that using a ~2B parameter model may not fully reflect how SWIFT performs at larger scales. To directly address this, we conducted additional experiments on recent, stronger base models — specifically, **Mistral-7B-v0.1** and **Qwen2.5-7B-Instruct** — using **Qwen3-32B** as the teacher. The results are reported below:
>
> |Setting|Model|Arc|Truthful|Wino|GSM8k|MMLU|HellaSwag|Avg|
> |-|-|-|-|-|-|-|-|-|
> ||Teacher|71.08|59.29|76.95|80.97|81.53|83.96|75.63|
> |Qwen3-32B→Mistral-7B-v0.1|Base|59.90|42.65|77.43|31.99|60.60|83.47|59.34|
> ||DPO|60.70|42.11|76.59|32.65|61.30|**83.92**|59.55|
> ||SPIN|61.73|44.63|77.07|33.14|62.76|81.68|60.17|
> ||SWIFT|**63.06**|**51.25**|**78.20**|**42.20**|**65.17**|81.13|**63.50**|
> |Qwen3-32B→Qwen2.5-7B-Instruct|Base|65.96|64.70|75.06|68.61|**73.49**|81.45|71.55|
> ||DPO|66.12|63.50|74.19|67.49|73.02|81.67|71.00|
> ||SPIN|66.53|65.76|75.17|69.35|72.38|82.30|71.92|
> ||SWIFT|**66.98**|**66.21**|**75.53**|**73.95**|73.36|**82.45**|**73.08**|
>
> These results confirm that SWIFT consistently improves performance, even when applied to larger and more capable models.
>
> ---
>
> **Response to Weakness3.**
>
> > 3. The proposed method heavily relies on previous work (self-play fine-tuning), which somewhat limits its technical contribution and novelty.
>
> **A3:** While our method builds upon the self-play fine-tuning framework (SPIN), it introduces key improvements that address important limitations in prior work.
>
> Specifically, SPIN applies uniform learning signals across all tokens, assuming that all tokens in a response are equally important. However, in real-world scenarios, ground-truth and generated responses often contain a mix of high- and low-quality tokens. Treating all tokens equally can amplify noise or misleading patterns during training. Our method addresses this by introducing token-level importance weights, estimated from a stronger teacher model, to guide learning more precisely.
>
> Beyond this practical improvement, we also provide a novel theoretical formulation of token-level self-play, along with complete proofs that ground our method in a rigorous framework. Additionally, we propose a robust and general solution for handling tokenizer mismatches between teacher and student models — a practical challenge often overlooked in knowledge distillation.
>
> Together, we believe these contributions represent a meaningful and practical advance over prior self-play fine-tuning methods, and offer both theoretical insights and empirical improvements.
>
> ---
>
> **Response to Question1.**
>
> > 1. In the proof of Lemma 3.1 in the Appendix, could you provide further clarification on why the existing proof is sufficient to establish the necessary and sufficient condition? Specifically, what steps or assumptions are being made that allow for this conclusion?
>
> **A1:** We are pleased to elaborate on the reasoning behind Lemma 3.1 and clarify why the proof is sufficient to establish the necessary and sufficient condition.
>
> We do not need to make any assumption for the proof of Lemma 3.1.
> Let us denote $A: P_d = P_{\theta^S}$ and $B: p_d(y_t \mid x, y_{<t}) = p_{\theta^S}(y_t \mid x, y_{<t}), ∀t$.
>
> Our proof in Lemma 3.1 proves $A \implies B$ by using induction. This means that $B$ is a necessary condition of $A$. We now provide the proof of $B \implies A$. We have
> $$p_{d}\left(x,y\right)=p_{d}(x)\prod_{i=1}^{T}p_{d}(y_{i}\mid x,y_{<i})$$
> $$p_{\theta^{S}}\left(x,y\right)=p_d(x)\prod_{i=1}^{T}p_{\theta^{S}}(y_{i}\mid x,y_{<i})$$
> Therefore, we reach $p_d(x,y) = p_{\theta^{S}}\left(x,y\right)$ or $P_d = P_{\theta^S}$. This implies that $B$ is a sufficient condition of $A$.
>
> ---
>
> **Response to Question2.**
>
> > 2. Could you elaborate on the relationship between the proposed approach and the theorems presented? A more explicit expression or explanation would help to clarify how these components interact and support the overall method.
>
> **A2:** We appreciate the opportunity to clarify how our theoretical framework supports the overall method and connects the key components of the proposed approach.
>
> First, our Lemma 3.1 indicates the connection between the distribution matching and token level distribution matching. This leads to the IPM (Integral Probabilistic Metric) formulas in Eq. (2). However, the reward functions engage in the IPM formulas in Eq. (2) are the token level reward functions $r_t([x,y_{<t}],y_t)$.
>
> We then propose the definition of consistency in Definition 3.2 to unify the token level reward functions to a unique setence level reward function $r(x,y_{\leq t})$. Furthermore, Theorem 3.3 allows us to equivalently transform the IPM form for the distribution matching regarding the sentence level reward function in Eq. (3). However, this follows the teacher-forcing paradigm. Therefore, in Theorem 3.4, we equivalently turn it to the student-forcing form in Eq. (5) or Eq. (6).
>
> We now connect the token level (sentence level) distribution matching viewpoint and the reinforcement viewpoint through the advantage functions in Eq. (7). Please read Lines from 167 to 173 in the main paper for the discussion of the connection indicating that Eq. (7) encourages the token level distribution matching, hence the sentence level distribution matching.
>
> We then develop Lemma 3.5 to see the connection between the optimal policy of Eq. (7) and the Q-value function (i.e., this is relevant to the reward function). Therefore, we can implicitly consider the reward function as an implicit function of the policy which is reflected in Eqs. (10) and (11). Fortunately, we can turn this connection to Eq. (12) in Lemma 3.6, leading to the trainable algorithm in Theorem 3.7 by refering to the optimization problem in (6).
>
> To summarize, our theoretical development connects the distribution matching viewpoint in Self-Play and the RL-based viewpoint through advantage functions which we believe opens doors for other ideas and developments.
>
> ---
>
> **Response to Question3.**
>
> > 3. While the experiments demonstrate improvements on small language models, it would be valuable to investigate the performance of the proposed approach on larger models. How does the method scale, and are there any limitations or challenges that arise when applying it to more complex models?
>
> **A3:** Please refer to our response to **Weakness2** for results on larger models. We find that SWIFT continues to provide clear performance gains even as the student model size increases. However, as with most knowledge distillation methods, using larger students often requires a strong enough teacher to guide the learning process. We recognize this limitation and plan to explore techniques for reducing teacher dependency in future work.
>
> ---
>
>
> **Response to Question4.**
>
> > 4. What is the computational efficiency of the proposed method compared to existing baselines? Are there any trade-offs between accuracy and computational resources, and if so, how do they impact the practicality of the approach?
>
> **A3:** The computational efficiency we mentioned in our paper is: In practice, unlike typical white-box knowledge distillation methods that require online access to the teacher model during training, which increases training time and memory usage, SWIFT performs a single offline forward pass of the teacher model over the training data to compute token-level importance weights. These weights are then saved and reused during training, eliminating the need to keep the teacher in memory. Furthermore, since each sample is independent, this step can be fully parallelized to significantly speed up computation.
>
> To further quantify this, we compare **training time per sample** and **peak GPU memory usage** across SFT, ULD [2], DSKD [1], and SWIFT. While we would like to note that our distillation codebase is based on DSKD [1], which uses a different training library than our SWIFT codebase, making direct comparisons challenging. However, we made every effort to ensure a fair comparison:
>
> | Method | Training Time / Sample (s) | Peak GPU Memory (GB) |
> |-|-|-|
> |SFT|0.26|51.08|
> |ULD|0.60|70.20|
> |DSKD|0.54|66.63|
> |SWIFT|0.30|55.39|
>
> These results show that SWIFT is comparable to SFT in runtime and memory usage, while more efficient than typical white-box distillation baselines.
>
> Because the teacher is only used once and the token weights are reused throughout training, we believe there is no trade-off between accuracy and computational resources in our approach.
>
> Once again, thank you for your comment and suggestion. If you have any additional comments or concerns, please feel free to share them with us.
>
> ---
>
> **References:**
>
> [1] Zhang, Songming, et al. "Dual-space knowledge distillation for large language models." arXiv preprint arXiv:2406.17328 (2024).
>
> [2] Boizard, Nicolas, et al. "Towards cross-tokenizer distillation: the universal logit distillation loss for llms." arXiv preprint arXiv:2402.12030 (2024).

---

> > ### Comment · Reviewer_ti82 · 2025-08-05
> >
> > Thank you for the response. The response has addressed my concerns. Please incorporate the rebuttal into the final version of your paper, especially the new experiments.

---

### Official Review · Reviewer_Qi3n · 2025-07-01

**Clarity:** 4
**Significance:** 3
**Originality:** 3
**Rating:** 4
**Confidence:** 3

**Summary:**

The paper proposes SWIFT (Self-Play Weighted Fine-Tuning)—a token-level, importance-aware extension of SPIN that leverages a stronger teacher model to estimate per-token rewards. The authors (i) derive the objective from an IPM formulation that links to advantage-maximization, (ii) introduce a simple tokenizer-mismatch mapping, and (iii) show that SWIFT can act both as a preference-alignment technique and as a lightweight knowledge-distillation method. On six alignment benchmarks for LLMs, small student LLMs gain significant more performance from teacher against other DPO-/KD-style baselines. Ablations confirm that teacher-guided weights dominate random/equal/contrastive alternatives.

**Questions:**

1. **Compute overhead.**  How many GPU-hours did the single offline teacher pass require for the 50k Ultrachat subset? Could you provide the computation cost of each distillation methods and the SFT baseline? Could it be more than pruning and fine-tuning the teacher model directly?
2. **Scaling.**  It seems that SWIFT gains the largest improvement in the first iteration, and could not consistently improve the student model in subsequent iterations (Fig 2). Could you provide some insights on the scaling potential of SWIFT along training iterations?
3. **Teacher quality dependence.**  Table 3 shows uniform weights still help, but less than the teacher to a large margin.  Can you quantify performance as a function of teacher–student capability gap and test with other teachers?
4. **Qualitative analysis.**  Can you provide some qualitative analysis on the trained models at each iteration? What domains do they improve on?

**Ethical Concerns:**

["NO or VERY MINOR ethics concerns only"]

**Final Justification:**

After reading the rebuttal and other reviews, I decide to raise my rating since the authors gave reasonable answers to my questions.

**Limitations:**

The paper lists teacher reuse overhead and future efficiency work, but should also discuss how to use this white-box distillation method practically. In real-world applications, the teacher model is often black-box and not easily accessible for analysis.

**Quality:**

3

**Strengths And Weaknesses:**

### Strengths

1. **Sound Object Derivation:** The authors provide detailed derivations of the SWIFT objective from an IPM perspective, linking it to advantage-maximization. The use of student forcing, optimal solutions, and the final form is simple to calculate.

2. **Practical Mitigation for Heteogeneous Tokenizers:** The paper introduces a straightforward method to handle tokenizer mismatches, which is a common issue in LLM distillation.

3. **Clear Experiment Settings:** The experiments are well-explained, under a white-box distillation setting, which is an important research problem.


### Weaknesses

1. **Limited Scope of Methodology:** The method is mainly extending the token-level DPO (ITS-DPO, etc.) to self-play settings. It is indeed an important extension, but its effectiveness is unclear for recent RL settings on LLMs, e.g., incentivizing long reasoning chains or agentic tasks. Current design and experimentation focus only on preference alignment.

2. **Inappropriate Baseline Comparisons:** There are a few settings differ between the baselines and SWIFT:
   (a) In the alignment experiments, SWIFT is trained on 50k SFT dataset and other DPO-style baselines are trained on 62k preference samples. This is not a fair comparison. At least SWIFT should be trained on 62k preference samples as well (using positive responses).
   (b) In the knowledge distillation experiments, only SFT/KL-based baselines are compared with SWIFT, while other DPO-style baselines are not included. For instance, by sampling from the student model, negative samples could be obtained for DPO-style baselines.
   (c) In Fig 1(b), the rewards calculated for SWIFT and SPIN are not in the same way. SWIFT does use a token-weighted reward. However, sample-level rewards in original DPO could also be obtained from the trained model, which is more direct opponent for SPIN's rewards. Different reward calculation methods could lead to unfair comparisons due to the scales of rewards might be different.

3. **Outdated Backbone Models:** The experiments are conducted on small student models from several years ago (e.g., GPT-2, Qwen-1.5). The effectiveness of the method on more recent instruction models (e.g., Gamma-3) or reasoning models (e.g., Qwen-3) is unclear. The paper should at least mention the limitations of the current experiments and the potential challenges of scaling to these models.

---

> ### Author Rebuttal · Authors · 2025-07-31
>
> Dear reviewer **Qi3n**, thank you for your recognition of our paper. We will now address your questions below.
>
> ---
> **Weakness1. Limited Scope of Methodology Experiments**
>
> A1: Thank you for pointing out the value of our method. Our original experiments followed the same setup as most recent DPO-style works, which mainly focus on preference alignment tasks. However, we understand your concern, so we conducted additional experiments focusing on reasoning and agentic tasks. The student model is **Qwen2.5-7B Instruct** and the teacher model is **Qwen3-32B**.
>
> For reasoning tasks, we use two challenging benchmarks: **Big-Bench-Hard (BBH)** and **Discrete Reasoning Over Paragraphs (DROP)**. The result in table below demonstrate that SWIFT consistently outperforms the baselines on reasoning-heavy tasks.
> ||Base|DPO|SPIN|SWIFT|
> |-|-|-|-|-|
> |BBH|65.07|64.23|64.85|**66.01**|
> |DROP|59.28|59.10|58.62|**62.03**|
>
> Regarding agentic tasks, we further evaluated on the **ToolBench** benchmark to assess SWIFT’s performance in agentic, tool interactions.
> |Metric|Base|DPO|SPIN|SWIFT|
> |-|-|-|-|-|
> |Act.EM(↑)|46.69|46.12|47.01|**48.36**|
> |F1(↑)|40.51|40.63|40.73|**42.08**|
> |HalluRate(↓)|1.60|1.44|1.42|**1.40**|
> |Rouge-L(↑)|4.35|4.57|4.40|**4.61**|
>
> These results support SWIFT’s applicability to complex reasoning task and real-world agentic tasks.
>
> ---
> **Weakness2. Concern on Baseline Comparisons: a) SWIFT uses a smaller 50k SFT dataset while DPO-style baselines use 62k preference samples; b) KD comparison lacks DPO-style baselines; c) reward formulas differ between SWIFT and SPIN in Fig 1(b)**
>
> A2:
> (a) We followed the same setup as the SPIN paper to make a fair comparison—using 50k SFT examples for SWIFT/SPIN and 62k preference examples for DPO-style methods. However, we agree that comparing on equal data sizes is also important. So, we ran an extra experiment using a larger 62k SFT dataset by adding random 12k new examples from UltraChat200k. When retrained on this larger SFT dataset, SWIFT achieved a final score on Open LLM Leaderboard of **47.52**, slightly above the original result of **47.46**, confirming that our findings are robust to dataset size differences.
>
> (b) Thank you for this suggestion. However, these DPO-style methods are originally designed for preference datasets, which are not really suitable with SFT datasets. Regarding your suggestion to simulate preference data by treating the ground-truth response as "chosen" and the model-generated response as "rejected", we believe this construction is suitable for the self-play mechanism of SWIFT or SPIN, and it is unclear whether such modifications would be effective for baselines like DPO or TIS-DPO. Due to rebuttal space constraints, we plan to incorporate a comparison with DPO-style methods adapted to the SFT data in the final version.
>
> (c) We appreciate your observation regarding Fig 1(b). This figure was designed to compare the actual reward signals used by SWIFT and SPIN during training rather than standardize rewards across all methods. Since SWIFT’s reward formulation is token-weighted, this plot reflects how the optimization differs in practice. However, we understand your concern and have generated a new figure where both methods use the same reward formula. Due to the rebuttal format limitations, we will include this new figure in the final version.
>
> ---
> **Weakness3. Limited Backbone Models**
>
> A3: We acknowledge that using student models with ~2B parameters may not fully reflect the scalability of our method. Therefore, we ran additional experiments on recent, stronger base models. These results in table below confirm that SWIFT continues to deliver meaningful performance gains even with newer and stronger instruction and reasoning models.
> |Setting|Model|Arc|Truthful|Wino|GSM8k|MMLU|HellaSwag|Avg|
> |-|-|-|-|-|-|-|-|-|
> ||Teacher|71.08|59.29|76.95|80.97|81.53|83.96|75.63|
> |Qwen3-32B→Mistral-7B-v0.1|Base|59.90|42.65|77.43|31.99|60.60|83.47|59.34|
> ||DPO|60.70|42.11|76.59|32.65|61.30|**83.92**|59.55|
> ||SPIN|61.73|44.63|77.07|33.14|62.76|81.68|60.17|
> ||SWIFT|**63.06**|**51.25**|**78.20**|**42.20**|**65.17**|81.13|**63.50**|
> |Qwen3-32B→Qwen2.5-7B-Instruct|Base|65.96|64.70|75.06|68.61|**73.49**|81.45|71.55|
> ||DPO|66.12|63.50|74.19|67.49|73.02|81.67|71.00|
> ||SPIN|66.53|65.76|75.17|69.35|72.38|82.30|71.92|
> ||SWIFT|**66.98**|**66.21**|**75.53**|**73.95**|73.36|**82.45**|**73.08**|
>
> We understand that, as in typical knowledge distillation settings, scaling up student models naturally increases dependency on suitably powerful teachers. We recognize this limitation and plan to explore in future work.
>
> ---
> **Question1. Compute overhead**
>
> A1: We report below the GPU-hours required for each phase of the SWIFT pipeline, measured on the 50k Ultrachat subset using a single NVIDIA H100 GPU.
> |Iteration|Generation|Compute Weight|Training|Teacher Forward|Overall|
> |-|-|-|-|-|-|
> |0|0.24h|0.58h|1.45h|0.19h|2.27h|
> |1|0.23h|0.57h|1.42h|0.19h|2.22h|
> |2|0.24h|0.58h|1.43h|0.19h|2.25h|
> |3|0.22h|0.55h|1.40h|0.18h|2.17h|
>
> The teacher forward pass only takes ~ 0.19 hours ~ 11 minutes for 50k samples, which we believe is quite efficient and does not contribute significantly to the overall training cost.
>
> We would like to note that our distillation codebase is based on DSKD, which uses a different training library than our SWIFT codebase, making direct comparisons challenging. However, we made every effort to fairly compare the **training time per sample** and **peak GPU memory usage** across SFT, ULD, DSKD, and SWIFT in the table below:
> |Method|TrainingTime/Sample(s)|Peak GPU Memory(GB)|
> |-|-|-|
> |SFT|0.26|51.08|
> |ULD|0.60|70.20|
> |DSKD|0.54|66.63|
> |SWIFT|0.30|55.39|
>
> These results show that SWIFT is comparable to SFT in runtime and memory usage, while more efficient than typical distillation baselines.
>
> Finally, we appreciate the reviewer’s suggestion to consider pruning as an alternative. While pruning can yield compact models, it often involves additional hyperparameter tuning and retraining overhead. However, we see it as a valuable idea and plan to consider it in future work.
>
> ---
> **Question2. Scaling iteration potential**
>
> A2: We believe most of SWIFT’s gains are in the first iteration because, at this stage, the student model is still weak and thus has more to learn. After that, the student already improves, so there’s less to fix, and learning slows down. Moreover, since SWIFT uses the student’s previous iteration to generate new responses. As the student gets better, it generates fewer mistakes and thus fewer learning opportunities.
>
> We acknowledge the limitation in scaling potential of SWIFT. However, while continued improvements slow over iterations, we see this fast convergence as an advantage—SWIFT achieves strong gains in few rounds, reducing training time.
>
> ---
> **Question3. Teacher quality dependence**
>
> A3: To better understand SWIFT's sensitivity to teacher quality, we ran more experiments using three different teacher models. We then reported the student model performance after training with each one:
> ||ARC|Truthful|Wino|GSM8k|MMLU|HellaSwag|Avg|
> |-|-|-|-|-|-|-|-|
> |zephyr-7b-sft-full|60.41|43.73|74.19|26.76|60.92|82.85|58.14|
> |zephyr-7b-sft-full→Qwen1.5-1.8B|39.78|39.12|61.48|37.93|44.84|61.63|47.46|
> |Qwen2.5-7B-Instruct|65.96|64.70|75.06|68.61|73.49|81.45|71.54|
> |Qwen2.5-7B-Instruct→Qwen1.5-1.8B|39.91|40.11|62.43|37.56|44.50|62.43|47.82|
> |Qwen3-32B|71.08|59.29|76.95|80.97|81.53|83.96|75.63|
> |Qwen3-32B→Qwen1.5-1.8B|40.27|38.96|63.51|38.83|44.69|62.19|48.07|
>
> These results show that while stronger teachers yield slightly better student performance, the gains are not strongly correlated with teacher capability. This is because the teacher is only used to estimate the importance weight of token, not to directly control what the model generates. As a result, SWIFT maintains stable and effective performance across diverse teacher–student pairs and does not depend heavily on highly capable teachers.
>
> ---
> **Question4. Qualitative analysis**
>
> A4: Following your suggestion, we conducted a qualitative analysis using the MMLU benchmark, which covers diverse knowledge domains. MMLU comprises four major sub-domains: Humanities, Social Sciences, STEM, and Other. The results are shown in the table below:
>
> |Domain|base|Ite 0|Ite 1|Ite 2|Ite 3|
> |-|-|-|-|-|-|
> |**Humanities**|39.85|**40.89**|40.13|40.70|40.78|
> |**SocialSci.**|47.43|48.29|47.90|48.88|**49.59**|
> |**STEM**|34.61|36.76|37.08|37.93|**38.12**|
> |**Other**|50.08|52.08|52.11|51.75|**52.14**|
>
> The results show clear and consistent improvements, especially in STEM and Social Sciences, which suggests the model is getting better at reasoning and logical thinking. Meanwhile, performance in Humanities and Other domains remains stable or improved, showing that the model’s reasoning gains didn’t hurt its general knowledge or overall quality.
>
> ---
> **Limitation. Practical use of the method**
>
> A: We appreciate the opportunity to clarify the practical workflow. In practice, SWIFT can be implemented efficiently by first running the teacher model in a single offline pass over the training data to estimate token-level importance weights. These weights are then saved with the dataset and simply loaded during training. Moreover, since each sample is independent, this step can be fully parallelized to significantly speed up computation. Notably, in the timing reported in our response to Question1, we only used sequential inference for simplicity; parallel inference would further reduce this time substantially.
>
> We agree that this approach assumes white-box access to the teacher, which may not hold in all settings. However, many recent studies indicate that white-box distillation often outperforms black-box. Moreover, the growing of powerful open-source models makes white-box distillation more feasible and desirable than ever. These trends motivate SWIFT to leverage the strengths of strong open teachers.

---

### Official Review · Reviewer_8XmY · 2025-07-03

**Clarity:** 2
**Significance:** 3
**Originality:** 2
**Rating:** 4
**Confidence:** 3

**Summary:**

This paper proposes SWIFT (Self-Play Weighted Fine-Tuning), a method to assign token-level importance weights from a strong teacher model, which are used to guide the student model via distillation. SWIFT is extended from a prior work SPIN, which lets the model to generate responses and improve them with human responses. While SPIN operates on the sequence level, SWIFT further applies token-level refinement via importance-aware token weighting. A general token mapping strategy is applied to solve the issue of unmatched tokenizers between student and teacher. Experiments on alignment setting and distillation setting show that the proposed method can improve from baseline methods across multiple benchmarks.

**Questions:**

- The performance improvement on Table 1 seems to be mainly from GSM8K compared with baselines. Is there an intuition why it's the case?

**Ethical Concerns:**

["NO or VERY MINOR ethics concerns only"]

**Final Justification:**

My rating does not change after rebuttal.

**Limitations:**

Limitations are discussed in Conclusion.

**Quality:**

3

**Strengths And Weaknesses:**

## Strengths
- The proposed method is intuitive and shows to be effective in experiments.
- The proposed approach of dealing with different student-teacher tokenizers is interesting.
- The comparison with SPIN shows the importance of token-level reweighting when training goes on, which justifies the effectiveness of the method.

## Weaknesses
- Although the paper is well-written in general, the presentation flow is a little disconnected. The theoretical analysis takes too much space, while the algorithm details and experiment results are not discussed extensively in the main paper. It would be helpful to emphasize the main theoretical results and insights, and discuss more on experiments.
- By introducing a teacher model, and computing token-level weight, the computational cost can be an issue. Is there a way to overcome it by reducing the computations? Or is it possible to combine SWIFT and SPIN?

---

> ### Author Rebuttal · Authors · 2025-07-31
>
> Dear reviewer 8XmY, we thank you for your review and for recognizing the intuition, novelty, and effectiveness of our approach. The following are our responses to each individual comment.
>
> ---
>
> **Response to Weakness1.**
> > Although the paper is well-written in general, the presentation flow is a little disconnected. The theoretical analysis takes too much space, while the algorithm details and experiment results are not discussed extensively in the main paper. It would be helpful to emphasize the main theoretical results and insights, and discuss more on experiments.
>
>
> **A1:**  Thank you for your suggestion regarding the balance between theoretical analysis and experimental details. We will carefully revise the final manuscript to emphasize the main theoretical results more clearly and allocate more space in the main paper for key algorithmic details and experimental results. We appreciate this feedback and believe these changes will improve the presentation and flow.
>
>
> ---
>
> **Response to Weakness2.**
> > By introducing a teacher model, and computing token-level weight, the computational cost can be an issue. Is there a way to overcome it by reducing the computations? Or is it possible to combine SWIFT and SPIN?
>
>
> **A2:** We agree that introducing a teacher model can add computational cost. However, compared to most recent knowledge distillation approaches—which often require the teacher to be used online throughout training and thus demand significant additional memory and compute—SWIFT only needs the teacher for a single forward pass to estimate token importance weights. In particular, SWIFT performs a single offline forward pass of the teacher model over the training data to compute token-level importance weights. These weights are then saved and reused during training, eliminating the need to keep the teacher in memory. Furthermore, since each sample is independent, this step can be fully parallelized to significantly speed up computation. This design keeps the overhead manageable and scalable.
>
> To quantify this, we report the GPU-hours required for each stage of the SWIFT pipeline on the 50k Ultrachat subset using a single NVIDIA H100 GPU. The main phases include: (1) response generation, (2) token weight computation (which includes the teacher forward pass), and (3) training. The breakdown per iteration is shown below:
>
> | Iteration | Generation (1) | Compute Weight (2) | Training (3) | Teacher Forward | Overall |
> |-----------|------------|----------------|----------|------------------|---------|
> | 0         | 0.24 h     | 0.58 h         | 1.45 h   | 0.19 h           | 2.27 h  |
> | 1         | 0.23 h     | 0.57 h         | 1.42 h   | 0.19 h           | 2.22 h  |
> | 2         | 0.24 h     | 0.58 h         | 1.43 h   | 0.19 h           | 2.25 h  |
> | 3         | 0.22 h     | 0.55 h         | 1.40 h   | 0.18 h           | 2.17 h  |
>
> As shown, the teacher forward pass (offline, single forward) only takes ~ 0.19 hours ~ 11 minutes for 50k samples, which we believe is quite efficient and does not contribute significantly to the overall training cost. Notably, in this table, we only used sequential (single-process) inference for simplicity; parallel inference would further reduce this time substantially.
>
> Furthermore, when using a teacher is impractical, we offer a teacher-free variant is contrastive weighting strategy, inspired by TIS-DPO [1]. As shown in Table 3 (ablation study), this approach achieves competitive performance without requiring any external teacher. Specifically, it trains separate “positive” and “negative” models via DPO, and estimates token weights based on the difference in their output logits. Because it does not rely on any teacher model, it remains a viable approach in scenarios where access to a strong teacher is unavailable, while still outperforming other non-teacher baselines. We believe this makes the method broadly applicable, even in resource-constrained settings.
>
> Finally, we view SWIFT as an advanced extension of SPIN, which can be flexibly reduced to SPIN or hybridized based on resource constraints. We will clarify this relationship in the final version.
>
> ---
>
> **Response to Question1.**
> > The performance improvement on Table 1 seems to be mainly from GSM8K compared with baselines. Is there an intuition why it's the case?
>
>
> **A3:** Thank you for noting the strong improvements on GSM8K. This benchmark is heavily focused on mathematical reasoning, where token-level signals are particularly valuable—rewarding intermediate reasoning steps, not just final answers. SWIFT’s fine-grained weighting helps the student focus on critical tokens, leading to more substantial gains. For other tasks, gains are also observed, but the token-level reward signal seems especially effective for the complex multi-step reasoning required by GSM8K.
>
>
> Once again, thank you for your thoughtful review and for recognizing the contributions of our work.
>
> ---
>
> **References:**
>
> [1]  Liu, Aiwei, et al. "Tis-dpo: Token-level importance sampling for direct preference optimization with estimated weights." arXiv preprint arXiv:2410.04350 (2024).

---

### Official Review · Reviewer_Qm7U · 2025-07-05

**Clarity:** 3
**Significance:** 2
**Originality:** 2
**Rating:** 4
**Confidence:** 4

**Summary:**

This paper proposes a method, namely, SWIFT (Self-Play Weighted Fine-Tuning), for aligning Large Language Models (LLMs) that improves upon existing self-play techniques. Existing methods apply a uniform learning signal across all tokens in a generated response, ignoring the fact that even "rejected" responses can contain many high-quality tokens, especially as the model improves. To address this, SWIFT introduces a fine-grained, token-level weighting mechanism into the self-play loop. Experiments are conducted in two settings: preference alignment and knowledge distillation.

**Questions:**

1. The teacher model provides the crucial token-level weights. If this teacher model has its own inherent biases (e.g., factual inaccuracies, stylistic tics, or harmful stereotypes), is there a risk that SWIFT will not only distill its capabilities but also amplify its flaws in the student? Does the proposed weighting scheme have any mechanism to mitigate the transfer of undesirable biases from the teacher?

2. The final objective in Theorem 3.7 is composed of two main terms, $u(⋅)$ and $v(⋅)$. The term $u(⋅)$ resembles the DPO objective applied to the advantage function. Could you provide a more intuitive explanation for the role of the second term, $v(⋅)$, which involves the difference of weighted sequence KL-divergences? How does it complement the first term to improve alignment?

3. The ablation study shows that teacher-guided weighting outperforms the contrastive weighting from TIS-DPO. Is this purely because the teacher model is stronger, or is there a fundamental difference in the kind of tokens that each method identifies as important? For example, does one method focus more on factual correctness while the other focuses on stylistic flair? A qualitative comparison of the highest-weighted tokens from each method would be insightful.

**Ethical Concerns:**

["NO or VERY MINOR ethics concerns only"]

**Final Justification:**

During rebuttal, authors adequately address my original questions. I maintain my original assessment.

**Limitations:**

The paper should more explicitly discuss the risk of error propagation in the iterative self-play loop, where a model trained on flawed self-generated data could degrade over time.

**Paper Formatting Concerns:**

No formatting issue.

**Quality:**

3

**Strengths And Weaknesses:**

Overall, this paper is easy to follow and well-motivated. The core contribution is to use a stronger "teacher" model to estimate the importance of each token in both the human-annotated ("chosen") and self-generated ("rejected") responses. Instead of traditional knowledge distillation that matches logits, SWIFT uses the teacher's outputs to assign reward-guided weights, effectively telling the student model which tokens to focus on during alignment.

Authors provide a solid theoretical foundation for this approach, deriving a token-level objective function from an Integral Probabilistic Metric (IPM) that connects to the advantage function used in reinforcement learning. It also introduces a practical method for transferring these token weights between models with different tokenizers by aligning them based on shared word segments.

Some weaknesses and potential improvements can be made:
1. The success of SWIFT is fundamentally tied to the availability of a powerful teacher model. The performance gains are contingent on the teacher providing high-quality importance signals. This dependency limits the method's applicability in scenarios where a significantly stronger, accessible teacher model does not exist. The ablation study confirms that without such guidance (e.g., using "equal weight"), the method's advantage diminishes.

2. The method relies on an iterative self-play process. While the results show consistent improvement, the paper does not fully explore the potential for error propagation. If the student model generates particularly poor responses in an early iteration, and the teacher's guidance is imperfect, the model could be trained on a flawed signal, which might be amplified in subsequent iterations. An analysis of the method's stability over many iterations or in lower-data regimes would be beneficial.

---

> ### Author Rebuttal · Authors · 2025-07-31
>
> Dear reviewer Qm7U, We’d like to thank you for acknowledging the novelty and the contribution of our proposed approach. We address your raised points as follows.
>
>
> ---
>
> **Weakness1. Dependency on a Strong Teacher Model for SWIFT’s Effectiveness**
>
> **A1:** We agree that a powerful teacher greatly benefits SWIFT by providing high‑quality token‑importance signals. Nevertheless, Table 3 (ablation study) shows that our contrastive‑weight strategy — inspired by TIS‑DPO [1] — achieves competitive performance without any external teacher. This variant relies solely on the base model: it trains separated “positive” and “negative” models via DPO and estimate token weights from their logits difference. Because it does not rely on any teacher model, it remains a viable approach in scenarios where access to a strong teacher is unavailable, while still outperforming other non-teacher baselines. We believe this makes the method broadly applicable, even in resource-constrained settings.
>
> ---
>
> **Weakness2. Risk of Error Propagation in Iterative Self-Play with Imperfect Guidance**
>
> **A2:** We agree that imperfect teacher guidance could adversely affect the student's training process. To directly address this concern, we conducted an additional robustness experiment in which we added random noise in the range [-0.2, 0.2] to the teacher’s token weights. As shown in the table below, performance dropped slightly, and SWIFT remained stable across iterations.
>
> | |Noise level|Arc|TruthfulQA|Winogrande|GSM8k|MMLU|HellaSwag|Average|
> |----------------------|-----------:|----:|-----------:|-----------:|-----:|-----:|----------:|--------:|
> |SWIFT (ite0)|0|39,16|41,01|61,96|33,43|44,41|61,69|46,94|
> |SWIFT (ite1)|0|38,91|38,93|61,80|37,91|43,83|61,32|47,12|
> |SWIFT (ite2)|0|39,76|40,02|62,04|33,60|44,61|62,07|47,02|
> |SWIFT (ite3)|0|39,78|39,12|61,48|37,93|44,84|61,63|47,46|
> |SWIFT (ite0)|±0.2|37,07|38,37|62,09|32,22|44,06|60,60|45,74|
> |SWIFT (ite1)|±0.2|37,58|38,95|61,75|34,51|43,45|60,85|46,18|
> |SWIFT (ite2)|±0.2|37,63|39,54|62,13|33,19|44,17|61,48|46,36|
> |SWIFT (ite3)|±0.2|38,06|39,08|62,01|35,18|44,09|61,53|46,66|
>
> Moreover, we would like to clarify why our method might actually have an advantage in lower-data regimes. A crucial premise of this paper is that ground truth can contain some low-reward tokens. These tokens can be understood as noise in SPIN [2]. By introducing token-level importance estimation, SWIFT can effectively identify and filter out such noisy tokens, optimizing primarily the high-quality parts of responses.
>
> ---
>
> **Question1. Risk of Amplifying Teacher Biases Through Token-Level Weighting**
>
> **A1:** We appreciate the concern that a teacher model’s own biases—whether factual, stylistic, or societal—could be inherited or even amplified. However, SWIFT incorporates two built‑in safeguards that inherently mitigate this risk:
>
> 1. **Clipped token weights:**
>    During the estimation of raw importance scores, we apply a clipping function to the teacher–student log-probability ratio, constraining token-level weights within the range $[L, U] = [-0.5, 1.5]$. This mechanism ensures that no single token can disproportionately influence training, significantly reducing the possibility of amplifying outlier biases or inaccuracies introduced by the teacher model.
>
> 2. **Smoothing via averaging token weights across shared word segments:**
>    When teacher and student models employ different tokenizers, we group tokens based on shared word-level components and then average token weights within each component. This method reduces the effect of any one token and makes the importance weights more balanced.
>
> We acknowledge that while these safeguards significantly mitigate potential bias transfer, they may not fully eliminate it. Thus, we consider this issue as a direction for future research and will clearly discuss these points in the final version.
>
> ---
>
> **Question2. Intuition Behind the $v(\cdot)$ Term in the SWIFT Objective**
>
> **A2:** To provide a clearer intuition, theorem 3.7 breaks the objective into two terms:
>
> - $u\left(x,y,y',\pi_{\theta^{S}},\omega\right)$ functions as a per-token alignment signal, similar in spirit to a token-level DPO update.
> - $v\left(x,y,y',\pi_{\theta^{S}},\omega\right) = \beta D_{SeqKL}\left(x,y,\omega;\pi_{\theta^{S}}\Vert\pi_{\theta_{k}^{S}}\right)-\beta D_{SeqKL}\left(x,y',\omega;\pi_{\theta^{S}}\Vert\pi_{\theta_{k}^{S}}\right)$
>  represents the difference between weighted sequence-level KL divergences from the updated student policy $\pi_{\theta^S}$ to its previous iteration $\pi_{\theta^S_k}$. It measures shifts in the model’s conditional distributions, carefully weighted by the token-level importance estimates.
>
> Intuitively, the $v(\cdot)$ term acts as a form of adaptive trust-region regularization. By penalizing excessive changes in the model’s conditional token distributions—particularly on high-importance tokens—it ensures stability and prevents the model from overfitting to narrow reward signals at the expense of general language quality or coherence. In this way, the term complements the alignment-driven updates from $u(\cdot)$ by encouraging conservative updates where necessary.
>
> Together, the two terms strike a balance: $u(\cdot)$ pushes the model to better align with preferred outputs, while $v(\cdot)$ tempers this push with a careful consideration of distributional drift, resulting in both effective and stable training dynamics.
>
> ---
>
> **Question3. Fundamental Differences Between Teacher-Guided and Contrastive Token Weighting**
>
> **A3:** We believe there is indeed a fundamental difference—not just a matter of teacher strength—in how each method identifies important tokens.
>
> Specifically, at each iteration, the contrastive weighting strategy adopted from TIS-DPO relies on two separate student models (a positive and a negative model), which are trained via DPO on pairs constructed from (ground_truth, model_response) and (model_response, ground_truth). Token importance is then estimated based on the difference in logits between these two models. While effective initially, this approach faces an inherent limitation as iterations progress: as the model improves over time, its generated responses become increasingly similar to ground truth. Consequently, the distinction between the ground truth and model response training pairs diminishes in later iterations, weakening the contrastive signal and thereby potentially reducing the quality of token importance estimates. In contrast, our proposed SWIFT method employs a fixed, external teacher model and it is unaffected by the student's iterative improvement, it provides stable, high-quality importance signals across iterations.
>
> Furthermore, we performed an additional qualitative analysis, comparing the top 20 highest-weighted tokens from each method. Token rankings were computed based on the average importance per appearance, defined by $\frac{\sum \text{importance weights assigned to this token}}{\text{number of appearances of this token}}$. The results are presented below:
>
> - **Top 20 tokens of SWIFT:**
>
> [`Serve`, `tering`, `antibiotic`, `intermitt`, `evaluates`, `destroys`, `Wealth`, `FDA`, `angelo`, `onne`, `visited`, `paralyzed`, `aken`, `toughest`, `Pricing`, `CLA`, `.dequeue`, `avenous`, `plings`, `specifies`]
>
> - **Top 20 tokens of TIS-DPO(D):**
>
> [`FACE`, `[@`, `LANG`, `.dest`, `ReturnValue`, `baseUrl`, `ĉPrint`, `keycode`, `-has`, `HG`, `repaint`, `Denver`, `FINAL`, `simil`, `ï¿½s`, `{/*`, `owler`, `.des`, `Teens`, `ÃĢ`]
>
> SWIFT selects semantically meaningful and contextually relevant tokens (e.g., "antibiotic", "FDA"), reflecting focus on factual and domain-rich content. In contrast, TIS-DPO often prioritizes tokens resembling code artifacts or noise (e.g., "ReturnValue", "[@"), suggesting less semantic alignment.
>
> However, we understand that our subjective judgment alone might not be sufficiently convincing, we further conducted an objective evaluation using GPT-4o as an external evaluator to assess  each method’s top-20 tokens. We designed the following targeted prompt for GPT-4o:
>
> ```text
> Below, I will provide the top 20 most important tokens across an entire dataset, as identified by two different methods. Please evaluate which method is better based on key criteria for token importance in a sentence, such as: semantic relevance (how meaningful or content-rich the token is), syntactic role (its grammatical contribution), contextual influence (how much it affects the surrounding words or sentence meaning), and task-specific utility (its contribution to downstream tasks like classification or retrieval)
> Here are these tokens:
> - **top_20_tokens_of_method1**:
>   {top_20_tokens_of_SWIFT}
> - **top_20_tokens_of_method2**:
>   {top_20_tokens_of_TIS-DPO}
> ```
>
> GPT-4o provided the following summarized judgment:
>
> ```text
> Method 1 is clearly better overall than Method 2.
> It includes more semantically rich, syntactically functional, and contextually influential tokens, many of which are likely to be task-relevant (e.g., medical, procedural, or evaluative terms).
> Method 2 appears to highlight code artifacts, UI labels, or tokenization noise, which are often less useful for language tasks.
> ```
>
> We hope this comprehensive analysis effectively addresses your question.
>
> ---
>
> **References:**
>
> [1] Liu, Aiwei, et al. "Tis-dpo: Token-level importance sampling for direct preference optimization with estimated weights." arXiv preprint arXiv:2410.04350 (2024).
>
> [2] Chen, Zixiang, et al. "Self-play fine-tuning converts weak language models to strong language models." arXiv preprint arXiv:2401.01335 (2024).

---

> ### Comment · Reviewer_Qm7U · 2025-08-05
>
> Thank you for your responses to my questions and the added experiments. After reading the authors' responses to other reviewers,I maintain my original assessment, and hope to see a revised and improved manuscript that incorporate reviewers' suggestions.

---

### Author Response · Authors · 2025-08-06
**General Response**

Dear Reviewers,

We sincerely appreciate your detailed comments and constructive suggestions, which have greatly contributed to improving our paper. Below, we summarize your feedback and our responses during the rebuttal process.

First, we are deeply encouraged by reviewers’ positive acknowledgments, specifically noting:

* The novelty and design of SWIFT are both clear and intuitive.

* Comprehensive theoretical foundations and detailed derivations linking our method to integral probabilistic metrics and reinforcement learning objectives.

* Extensive experimentation across diverse benchmarks and settings demonstrates consistent and significant performance improvements.

* SWIFT also introduces an effective and interesting method to address the tokenizer differences between teacher and student models.

During the rebuttal, we made our best efforts to provide a detailed response to every question from the reviewers. We have supplemented all required experiments and analyses that address all the reviewers’ concerns, including:

* **Robustness:** To explore the robustness of SWIFT, we injected random noise into token weights (±0.2) estimated by teacher model and showed that SWIFT maintains stable performance across iterations, underscoring the method’s stability.

* **Reasoning and agentic tasks evaluation:** Additional experiments show that SWIFT outperforms DPO and SPIN on reasoning (BBH, DROP) and agentic tasks (ToolBench), confirming its broader applicability.

* **Qualitative comparison of top tokens:** We compared top-weighted tokens from SWIFT and TIS-DPO, and external evaluation (GPT-4o) confirmed SWIFT selects more meaningful and relevant tokens.

* **Qualitative analysis on the trained models at each iteration:** We provided per-iteration qualitative analyses across multiple domains in MMLU benchmark, clearly demonstrating that improvements are primarily driven by enhanced reasoning capabilities, particularly in **STEM** and **Social Sciences**, without sacrificing general knowledge retention.

* **Computational efficiency:** We reported detailed GPU-hour breakdowns. A single offline teacher pass takes only ~ 11 minutes for 50k samples, which we consider efficient and does not significantly contribute to the overall training cost per iteration (~ 2 hours 13 minutes). Furthermore, **training time** and **peak GPU memory** remain comparable to standard SFT and are more efficient than typical distillation baselines.

* **Scaling model sizes:** We extended our experiments by scaling up student model and teacher model sizes. In all cases, SWIFT continues to deliver consistent and robust gains, demonstrating strong scalability

* **Teacher quality dependence.** We conducted further experiments evaluating SWIFT across different teacher-student capability gaps. Results show that SWIFT maintains stable performance regardless of teacher strength, highlighting robustness and broad applicability.

Furthermore, we have carefully addressed all reviewers' questions, providing detailed theoretical explanations, clarifying misunderstandings, outlining the practical workflow, which we believe significantly enhances the clarity and contribution of our method.

The above responses, experiments and analyses firmly confirm the effectiveness and advancement of our method. Based on the reviewers' final comments, we are pleased that the concerns and misunderstandings have been satisfactorily addressed. As promised, all these improvements will be carefully incorporated into the final version.

Best regards,

Authors

---

### Decision · Program_Chairs · 2025-09-17

**Decision:**

Accept (poster)

**Comment:**

(a) Summary:
We propose SWIFT (Self-Play Weighted Fine-Tuning), a token-level extension of self-play methods for aligning large language models (LLMs). Unlike prior approaches that apply uniform rewards across tokens, SWIFT leverages a stronger teacher model to estimate per-token importance, allowing useful supervision even from partially rejected responses. Building on SPIN, SWIFT introduces a token-weighted objective and a practical tokenizer-mapping strategy for student–teacher mismatch. Experiments in both preference alignment and knowledge distillation show consistent gains over DPO- and KD-style baselines across six benchmarks. Ablations further highlight the advantage of teacher-guided token weighting over uniform or heuristic alternatives.


(b) Strengths:

1. The paper provides a solid theoretical foundation and proposes a practical method.

2. The baselines and ablation study in the experiments are comprehensive.

(c) Weaknesses (after rebuttal):
1. Some writing issues exist and the presentation flow is a little disconnected.

2. Introducing a teacher model can add computational cost.

(d) Why this decision:
The paper is on the borderline. The rebuttal has done a good job by addressing reviewers' concerns in the first round of review. All reviewers vote for borderline accept consistently.

(e) Summary of discussions:
See above